# EMERGENCE OF EQUIVARIANCE IN DEEP ENSEMBLES

## ABSTRACT

We demonstrate that a generic deep ensemble is emergently equivariant under data augmentation in the large width limit. Specifically, the ensemble is equivariant at any training step for any choice of architecture, provided that data augmentation is used. This equivariance also holds off-manifold and is emergent in the sense that predictions of individual ensemble members are not equivariant but their collective prediction is. As such, the deep ensemble is indistinguishable from a manifestly equivariant predictor. We prove this theoretically using neural tangent kernel theory and verify our theoretical insights using detailed numerical experiments.

## 1 INTRODUCTION

Many machine learning tasks feature important symmetry constraints. As an example, predicting the energy of a molecule should not depend on the global orientation and position used to represent the molecule. This motivates the development of equivariant machine learning models which take the symmetry of the learning problem into account. In particular, manifestly equivariant deep learning architectures have for several reasons received significant attention in recent years: first, the inductive bias induced by manifest equivariance simplifies the learning problem (Bronstein et al., 2021; Elesedy & Zaidi, 2021). Second, built-in equivariance often leads to more robust predictions (Müller et al., 2021; Liu et al., 2022). Third, machine learning is increasingly deployed in the natural sciences for which equivariance is a central ingredient (Atz et al., 2021; Unke et al., 2021; Cranmer et al., 2023). A downside of manifestly equivariant models is however that they have a necessarily restricted architecture which has to be specifically designed for the symmetry properties of the problem at hand (Schütt et al., 2021; Batzner et al., 2022; Bacchio et al., 2023) and can be computationally costly (Cobb et al., 2021; Puny et al., 2021b). An alternative is to train a non-equivariant model with data augmentation and thereby learn the symmetry properties instead of incorporating them as a constraint into the architecture.

Similarly, deep ensembles (Lakshminarayanan et al., 2017), which average the predictions of several models trained from different initializations, are an important part of the deep learning toolbox. They have gained popularity as a method to estimate uncertainties of neural network predictions (Abdar et al., 2021; Linander et al., 2022) and have furthermore been shown to be more robust and to lead to an increase in performance compared to individual models (Ganaie et al., 2022).

In this paper, we show theoretically that deep ensembles are closely related to equivariant models. Specifically, we show that upon full data augmentation, deep ensembles become equivariant *at all training steps* in the large width limit. This statement even holds off-manifold and at initialization. Intuitively this can be understood as follows: at initialization, the deep ensemble predicts the same output for all inputs and is therefore in particular equivariant. Due to data augmentation, the deep ensemble is then trained in such a way that it stays equivariant. It is important to emphasize that this manifest equivariance is emergent: while the prediction of the ensemble is equivariant, the predictions of its members are not. In particular, the ensemble members are not required to have an equivariant architecture.

We rigorously derive this surprising emergent equivariance by using the duality between neural networks and kernel machines in the large width limit (Neal, 1996; Lee et al., 2018; Yang, 2020). The neural tangent kernel (NTK) describes the evolution of deep neural networks during training (Jacot et al., 2018). In the limit of infinite width, the neural tangent kernel is frozen, i.e., it does not evolve during training and the training dynamics can be solved analytically. As a random variable over initializations, the output of the neural network after arbitrary training time follows a Gaus-

sian distribution whose mean and covariance are available as closed form expressions (Lee et al., 2019). In this context, deep ensembles can be interpreted as a Monte-Carlo estimate of the corresponding expected network output. This insight allows us to theoretically analyze the effect of data augmentation throughout training and show that the deep ensemble is fully equivariant.

In practice, this emergent equivariance of deep ensemble cannot be expected to hold perfectly and exact equivariance will be broken, since the real-world neural networks are not infinitely wide and the expectation value over initalizations is estimated by Monte-Carlo. Furthermore, in the case of a continuous symmetry group, data augmentation cannot cover the entire group orbit and is thus approximate. We analyze the resulting breaking of equivariance and demonstrate in detailed numerical experiments that the deep ensembles nevertheless show a high degree of equivariance even with a low number of ensemble members.

The main contributions of our work are:

- We prove that infinitely wide deep ensembles are equivariant at all stages of training if trained with full data augmentation using the theory of neural tangent kernels.

- We carefully analyze the limitations of our theoretical analysis. Specifically, we derive bounds for deviations from equivariance due to finite size as well as data augmentation for a continuous group.

- We demonstrate the emergent equivariance in three different settings: the Ising model, FashionMNIST, and a high-dimensional medical dataset of histological slices.

## 2 RELATED WORKS

Deep ensembles were introduced by Lakshminarayanan et al. (2017) for uncertainty estimation and have been applied in many different contexts, a review is given in Ganaie et al. (2022).

There is a large body of literature on equivariant neural networks, reflecting the importance of the topic in recent years. The first group equivariant convolution was introduced in Cohen & Welling (2016), a more mathematically focused review is given in Gerken et al. (2023). That linear equivariant models benefit from equivariance was shown rigorously in Elesedy & Zaidi (2021), the universality of equivariant point-cloud networks was studied in Dym & Maron (2020). Ensembles of networks with different symmetry properties were studied in Loh et al. (2023). The relation between manifest equivariance and data augmentation with regards to model performance for invariant and equivariant tasks was studied in Gerken et al. (2022) and with regards to training dynamics in Flinth & Ohlsson (2023). Equivariance can also be achieved without constraining to a equivariant architecture by transforming the input by symmetrization over (an appropriately chosen subset of) the group orbit Puny et al. (2021a); Basu et al. (2023a;b) or a canonicalization network Kaba et al. (2023); Mondal et al. (2023). This approach is orthogonal to ours: instead of an ensemble of models and ensemble of inputs is considered. Note that the memory footprint of the symmetrization depends on the size of the group orbit while, for deep ensembles, it depends on the number of ensemble members. On the other hand, canonicalization and symmetrization leads to exact equivariance while deep ensembles naturally allow for uncertainty estimation and increased robustness.

Wide neural networks have been studied for a long time. That Bayesian neural networks behave as Gaussian processes was first discovered in Neal (1996), this result was extended to deep neural networks in Lee et al. (2018). Neural tangent kernels (NTKs), which capture the evolution of wide neural networks under gradient descent training, were introduced in Jacot et al. (2018). The literature on this topic has since expanded considerably so that we can only cite some selected works, a review on the topic is given in Golikov et al. (2022). The NTK for CNNs was computed in Arora et al. (2019). Lee et al. (2019) used the NTK to show that wide neural networks trained with gradient descent become Gaussian processes and Yang (2020) introduced a comprehensive framework to study scaling limits of wide neural networks rigorously. This framework was used in Yang & Hu (2022) to find a parametrization suitable for scaling networks to large width. NTKs were used to study GANs Franceschi et al. (2022), PINNs Wang et al. (2022), backdoor attacks Hayase & Oh (2022) as well as pruning Yang & Wang (2023), amongst other applications. Corrections to the infinite-width limit in particular in connection to quantum field theory have been investigated as well (Huang & Yau, 2020; Yaida, 2020; Halverson et al., 2021; Erbin et al., 2022).

With Novak et al. (2020), a Python package is available which automatizes the computation and evaluation of the NTK for many common network architectures.

Data augmentation has been studied in the context of kernel machines in some works. In particular, Mroueh et al. (2015), Raj et al. (2017) and Mei et al. (2021) study properties of kernel machines using group-averaged kernels but they do not consider wide neural networks. Dao et al. (2019) use a Markov process to model random data augmentations and show that an optimal Bayes classifier in this context becomes a kernel machine. This paper also shows that training on augmented data is equivalent to using an augmented kernel. Li et al. (2019) introduce new forms of pooling to improve kernel machines. As part of their analysis, they derive the analogous augmented kernel results as Dao et al. (2019) for the NTK at infinite training time. In contrast, we focus on the symmetry properties of the resulting (deep) ensemble of infinitely wide neural networks. In particular, we analyze the behavior of the ensemble at finite training time, show that their assumption of an "equivariant kernel" is satisfied for any orthogonal representation (cf. Theorem 2), include equivariance on top of invariance and derive a bound for the invariance error accrued by approximating a continuous group with finitely many samples.

## 3 DEEP ENSEMBLES AND NEURAL TANGENT KERNELS

In this section, we give a brief overview over deep ensembles and their connection to NTKs.

**Deep Ensemble:** Let $f_w : X \to \mathbb{R}$ be a neural network with parameters $w$ which are initialized by sampling from the density $p$, i.e. $w \sim p$. For notational simplicity, we consider only scalar-valued networks in the main part of the paper unless stated otherwise. Our results however hold also for vector-valued networks. The output of the deep ensemble $\bar{f}_t$ of the network $f_w$ is then defined as the expected value over initializations of the trained ensemble members

$$\bar{f}_t(x) = \mathbb{E}_{w \sim p} \left[ f_{\mathcal{L}_t w}(x) \right] , \tag{1}$$

where the operator $\mathcal{L}_t$ maps the initial weight $w$ to its corresponding value after $t$ steps of gradient descent. In practice, the deep ensemble is approximated by a Monte-Carlo estimate of the expectation value using a finite number $M$ of initializations

$$\bar{f}_t(x) \approx \hat{f}_t(x) = \frac{1}{M} \sum_{i=1}^{M} f_{\mathcal{L}_t w_i}(x), \quad \text{where} \quad w_i \sim p . \tag{2}$$

This amounts to performing $M$ training runs with different initializations and averaging the outputs of the resulting models. It is worthwhile to note that in the literature, the average $\hat{f}_t$ as defined in (2) is often referred to as the deep ensemble (Lakshminarayanan et al., 2017). In this work, we will however use the term deep ensemble to refer to the expectation value $\bar{f}_t$ of (1). Analogously, we refer to $\hat{f}_t$ as the MC estimate of the deep ensemble $\bar{f}_t$.

**Relation to NTK:** In the infinite width limit, a deep ensemble follows a Gaussian distribution described by the neural tangent kernel (Jacot et al., 2018)

$$\Theta(x, x') = \sum_{l=1}^{L} \mathbb{E}_{w \sim p} \left[ \left( \frac{\partial f_w(x)}{\partial w^{(l)}} \right)^{\top} \frac{\partial f_w(x')}{\partial w^{(l)}} \right] , \tag{3}$$

where $w^{(l)}$ denotes the parameters of the $l^{\text{th}}$ layer and we have assumed that the network has a total of $L$ layers. In the following, we use the notation

$$\Theta_{ij} = \Theta(x_i, x_j) \tag{4}$$

for the Gram matrix, i.e. the kernel evaluated on two elements $x_i$ and $x_j$ of the training set

$$\mathcal{T} = \{(x_i, y_i) \,|\, i = 1, \dots, |\mathcal{T}|\} . \tag{5}$$

Using the NTK, we can analytically calculate the distribution of ensemble members in the large width limit for a given input $x$ at any training time $t$ for learning rate $\eta$: The trained networks follow

a Gaussian process distribution with mean function $\mu_t$ and covariance function $\Sigma_t$ which are given in terms of the NTK by (Lee et al., 2019)

$$\mu_t(x) = \Theta(x, x_i) \left[ \Theta^{-1} \left( \mathbb{I} - \exp(-\eta \Theta t) \right) \right]_{ij} y_j \, , \tag{6}$$

$$\Sigma_t(x, x') = \mathcal{K}(x, x') + \Sigma_t^{(1)}(x, x') - (\Sigma_t^{(2)}(x, x') + \text{h.c.}) \, , \tag{7}$$

where all sums over the training set are implicit by the Einstein summation convention and we have defined

$$\Sigma_t^{(1)}(x, x') = \Theta(x, x_i) \left[ \Theta^{-1} \left( \mathbb{I} - \exp(-\eta \Theta t) \right) \mathcal{K} \left( \mathbb{I} - \exp(-\eta \Theta t) \right) \Theta^{-1} \right]_{ij} \Theta(x_j, x') \, , \tag{8}$$

$$\Sigma_t^{(2)}(x, x') = \Theta(x, x_i) \left[ \Theta^{-1} \left( \mathbb{I} - \exp(-\eta \Theta t) \right) \right]_{ij} \mathcal{K}(x_j, x') \, , \tag{9}$$

with the NNGP kernel

$$\mathcal{K}(x, x') = \mathbb{E}_{w \sim p} \left[ f_w(x) \, f_w(x') \right] \, . \tag{10}$$

The Gram matrix of the NNGP is given by $\mathcal{K}_{ij} = \mathcal{K}(x_i, x_j)$.

**Remark 1.** *This implies in particular that the ensemble output is given by $\bar{f}_t(x) = \mu_t(x)$ and ensemble members have output variance $\Sigma_t(x) := \Sigma_t(x, x)$ across initializations.*

In practice, the cost of inverting the Gram matrix is prohibitive. Therefore, one typically estimates the deep ensemble by (2) using $M$ trained models with different random initalizations. Nevertheless, the dual NTK description allows us to reason about the properties of the exact deep ensemble. In the following, we will use this duality to theoretically investigate the effect of data augmentation on the deep ensemble.

## 4    EQUIVARIANCE AND DATA AUGMENTATION

In this section, we summarize basics facts about representations of groups, equivariance, and data augmentation and establish our notation.

**Representations of Groups**    Groups abstractly describe symmetry transformations. In order to describe how a group transforms a vector, we use group representations. A (linear) representation of a group $G$ is a map $\rho : G \to \mathrm{GL}(V)$ where $V$ is a vector space and $\rho$ is a group homomorphism, i.e. $\rho(g_1)\rho(g_2) = \rho(g_1 g_2)$ for all $g_1, g_2 \in G$. Of particular importance are orthogonal representations for which $\rho(g^{-1}) = \rho(g)^\top$. In other words, representations that have orthogonal representation matrices. We will focus on these representations in the following. Importantly, this is a mild restriction which is satisfied for most important cases like rotations or permutations on finite-dimensional vector spaces.

**Equivariance**    For learning tasks in which data $x$ and labels $y$ transform under group representations, the map $x \mapsto y$ has to be compatible with the symmetry group; this property is called equivariance. Formally, let $f : X \to Y$ denote a (possibly vector valued) model with input space $X$ and output space $Y$ on which the group $G$ acts with representations $\rho_X$ and $\rho_Y$, respectively. Then, $f$ is equivariant with respect to the representations $\rho_X$ and $\rho_Y$ if it obeys

$$\rho_Y(g)f(x) = f(\rho_X(g)x) \qquad \forall x \in X, g \in G \, . \tag{11}$$

Similarly, a model $f$ is invariant with respect to the representation $\rho_X$ if it satisfies the above relation with $\rho_Y$ being the trivial representation, i.e. $\rho_Y(g) = \mathbb{I}$ for all $g \in G$. Considerable work has been done to construct manifestly equivariant neural networks with respect to specific, practically important special cases of (11). It has been shown both empirically (e.g. in Thomas et al. (2018), Bekkers et al. (2018)) and theoretically (e.g. in Sannai et al. (2021), Elesedy & Zaidi (2021)) that equivariance can lead to better sample efficiency, improved training speed and greater robustness. A downside of equivariant architectures is that they need to be purpose-built for symmetry properties of the problem at hand since standard well-established architectures are mostly not equivariant.

**Data Augmentation**    An alternative approach to incorporate information about the symmetries of the data into the model is data augmentation. Instead of using the original training set $\mathcal{T}$, we use a set which is augmented by all elements of the group orbit, i.e.

$$\mathcal{T}_{\text{aug}} = \{(\rho_X(g)x, \rho_Y(g)y) | g \in G, (x, y) \in \mathcal{T}\} \, . \tag{12}$$

In stochastic gradient descent, we randomly draw a minibatch from this augmented training set to estimate the gradient of the loss. If the group has finite order, data augmentation has the immediate consequence that the action of any group element $g \in G$ on a training sample can be written as a permutation $\pi_g$ of the indices of the augmented training set $\mathcal{T}_{\text{aug}}$, i.e.

$$\rho_X(g)x_i = x_{\pi_g(i)} \qquad \text{and} \qquad \rho_Y(g)y_i = y_{\pi_g(i)}, \tag{13}$$

where $i \in \{1, \ldots, |\mathcal{T}_{\text{aug}}|\}$. Data augmentation has the advantage that it does not impose any restrictions on the architecture and is hence straightforward to implement. However, the symmetry is only learned and it can thus be expected that the model is only (approximately) equivariant towards the end of training and on the data manifold. Furthermore, the model cannot benefit from the restricted function space which the symmetry constraint specifies.

## 5 EMERGENT EQUIVARIANCE FOR LARGE-WIDTH DEEP ENSEMBLES

In this section, we prove that any large-width deep ensemble is emergently equivariant when data augmentation is used. We refer to Appendix A for complete proofs. After stating our assumptions, the sketch the proof in three steps.

**Assumptions** We consider networks $f_w$ with parameters $w$. Crucially, we do not assume any equivariance properties of these networks. Furthermore, we require that the networks $f_w$ depend on their input $x$ only through expressions of the form $w^{(k)}x$ with $w^{(k)}$ a trainable matrix with components initialized from a centered Gaussian distribution. We emphasize that this condition is very mild and is satisfied for almost all common network architectures such as MLPs, CNNs, ResNets or transformers. It would be violated only in exotic scenarios such as a network with a skip connection from the input directly to the output.

We consider a finite group $G$ with orthogonal representations $\rho_X$ and $\rho_Y$ as well as data augmentation with respect to these representations, as discussed above. The case of continuous groups will be discussed subsequently.

**Step 1:** The representation $\rho_X$ acting on the input space $X$ induces a canonical transformation of the NTK and NNGP kernel:

$$\Theta(x, x') \quad \rightarrow \quad \Theta(\rho_X(g)x, \rho_X(g)x'). \tag{14}$$

$$\mathcal{K}(x, x') \quad \rightarrow \quad \mathcal{K}(\rho_X(g)x, \rho_X(g)x'). \tag{15}$$

For any orthogonal representation $\rho_X$ acting on the input space $X$, this canonical transformation leaves the kernels invariant:

**Theorem 2** (Kernel invariance under orthogonal representations). *Let $G$ be a group and $\rho_X$ an orthogonal representation of $G$ acting on the input space $X$. Under the representation $\rho_X$, the neural tangent kernel $\Theta$ as defined in (3) as well as the NNGP kernel $\mathcal{K}$ as defined in (10) of a neural network satisfying the assumptions above are invariant:*

$$\Theta(x, x') = \Theta(\rho_X(g)x, \rho_X(g)x'), \tag{16}$$

$$\mathcal{K}(x, x') = \mathcal{K}(\rho_X(g)x, \rho_X(g)x'). \tag{17}$$

*for all $g \in G$ and $x, x' \in X$.*

*Proof.* See Appendix A.

While this kernel invariance is shared by many standard kernels, such as RBF or linear kernels, this property is non-trivial for NTK and NNGP since they are not simply functions of the norm of the difference or inner product of the two input values $x$ and $x'$. Furthermore, this result holds irrespective of whether a group is of finite or infinite order.

**Step 2:** Data augmentation allows to rewrite the group action as a permutation, see (13). Combining this with the invariance of the kernels under orthogonal representations, we can shift a permutation from the first to the second index of the Gram matrix, i.e.,

$$\Theta_{\pi_g(i),j} = \Theta_{i,\pi_g^{-1}(j)}. \tag{18}$$

This statement can be easily shown as follows:

$$\Theta(x_{\pi_g(i)}, x_j) = \Theta(\rho_X(g)x_i, x_j) = \Theta(x_i, \rho_X(g)^{-1}x_j) = \Theta(x_i, x_{\pi_g^{-1}(j)}),  \tag{19}$$

where we have used the Theorem 2 derived in the last step for the second equality. Equation 18 then follows by the definition of the Gram matrix $\Theta_{ij} = \Theta(x_i, x_j)$. Along similar lines, the following result can be derived:

**Lemma 3** (Shift of permutation). *Data augmentation implies that one can shift the permutation group action from the first index to the second index for any matrix-valued analytical function $F$ involving the Gram matrices of the NNGP and NTK as well as their inverses:*

$$F(\Theta, \Theta^{-1}, \mathcal{K}, \mathcal{K}^{-1})_{\pi_g(i),j} = F(\Theta, \Theta^{-1}, \mathcal{K}, \mathcal{K}^{-1})_{i,\pi_g^{-1}(j)}.  \tag{20}$$

*where $\pi_g$ denotes the group action in terms of training set permutations, see (13).*

**Step 3:** Using Lemma 3, it can be shown that the deep ensemble is equivariant in the infinite width limit. Before stating the general theorem, we first illustrate the underlying reasoning by showing one particular consequence, i.e., that the mean is invariant if the output space $Y$ is equipped with the trivial representation $\rho_Y(g) = \mathbb{I}$. By (6), the output of the deep ensemble for transformed input $x \to \rho_X(g)x$ is given by

$$\bar{f}_t(\rho_X(g)\,x) = \mu_t(\rho_X(g)\,x) = \Theta(\rho_X(g)\,x, x_i) \left[\Theta^{-1}(\mathbb{I} - \exp(-\eta\Theta t))\right]_{ij} y_j  \tag{21}$$

Using Theorem 2, we can rewrite this as

$$\bar{f}_t(\rho_X(g)\,x) = \Theta(x, \rho_X(g)^{-1}\,x_i) \left[\Theta^{-1}(\mathbb{I} - \exp(-\eta\Theta t))\right]_{ij} y_j  \tag{22}$$

$$= \Theta(x, x_{\pi_g^{-1}(i)}) \left[\Theta^{-1}(\mathbb{I} - \exp(-\eta\Theta t))\right]_{ij} y_j  \tag{23}$$

$$= \Theta(x, x_i) \left[\Theta^{-1}(\mathbb{I} - \exp(-\eta\Theta t))\right]_{\pi_g(i)j} y_j,  \tag{24}$$

where we have changed the summation index $i \to \pi_g(i)$ in the last step. Using Lemma 3 and an analogous change in summation index, we can shift the permutation on the labels $y$:

$$\bar{f}_t(\rho_X(g)\,x) = \Theta(x, x_i) \left[\Theta^{-1}(\mathbb{I} - \exp(-\eta\Theta t))\right]_{ij} y_{\pi_g(j)}  \tag{25}$$

Since the output representation $\rho_Y$ is trivial by assumption, the outputs are invariant $y_{\pi_g(k)} = y_k$. It then immediately follows that the ensemble is invariant as well:

$$\bar{f}_t(\rho_X(g)\,x) = \bar{f}_t(x).  \tag{26}$$

Using analogous reasoning, the following more general result can be derived:

**Theorem 4** (Emergent Equivariance of Deep Ensembles). *The distribution of (possibly vector-valued) large-width ensemble members $f_w : X \to Y$ is equivariant with respect to the orthogonal representations $\rho_X$ and $\rho_Y$ of the group $G$ if data augmentation is applied. In particular, the ensemble is equivariant,*

$$\bar{f}_t(\rho_X(g)\,x) = \rho_Y(g)\,\bar{f}_t(x)  \tag{27}$$

*for all $g \in G$. This result holds*

1. *at any training time $t$,*

2. *for any element of the input space $x \in X$.*

We stress that this results holds even off the data manifold, i.e., for out-of-distribution data, and in the early stages of training as well as at initialization. As a result, it is not a trivial consequence of the training. Furthermore, we do not need to make any restrictions on the architectures of the ensemble members. In particular, the individual members will generically not be equivariant. However, their averaged prediction will be (at least in the large width limit). In this sense, the equivariance is emergent.

## 6 LIMITATIONS: APPROXIMATE EQUIVARIANCE

In the following, we discuss the breaking of equivariance due to i) statistical fluctuations of the estimator due the finite number of ensemble members, ii) continuous symmetry groups which do not allow for complete data augmentation, and iii) finite width corrections in NTK theory.

**Finite Number of Ensemble Members**  We derive the following bound for estimates of deep ensembles in the infinite width limit:

**Lemma 5** (Bound for finite ensemble members). *The deep ensemble $\bar{f}_t$ and its estimate $\hat{f}_t$ do not differ by more than threshold $\delta$,*

$$|\bar{f}_t(x) - \hat{f}_t(x)| < \delta\,, \tag{28}$$

*with probability $1 - \epsilon$ for ensemble sizes $M$ that obey*

$$M > -\frac{2\Sigma_t(x)}{\delta^2} \ln\left(\sqrt{\pi}\epsilon\right)\,. \tag{29}$$

We stress that the covariance $\Sigma$ is known in closed form, see (7). As such, the right-hand-side can be calculated exactly. We note that we also derive a somewhat tighter bound in Appendix A which however necessitates to numerically solve for $M$.

**Continuous Groups**  For a continuous group $G$, consider a finite subgroup $A \subset G$ which is used for data augmentation. We quantify the discretization error of using $A$ instead of $G$ by

$$\epsilon = \max_{g \in G} \min_{g' \in A} ||\rho_X(g) - \rho_X(g')||\,. \tag{30}$$

Then, the invariance error of the mean (6) is bounded by $\epsilon$:

**Lemma 6** (Bound for continuous groups). *Consider a deep ensemble of neural networks with Lipschitz continuous derivatives with respect to the parameters. For an approximation $A \subset G$ of a continuous symmetry group $G$ with discretization error $\epsilon$, the prediction of the ensemble trained on $A$ deviates from invariance by*

$$|\bar{f}_t(x) - \bar{f}_t(\rho_X(g)\,x)| \le \epsilon\, C(x)\,, \qquad \forall g \in G\,,$$

*where $C$ is independent of $g$.*

**Finite Width**  Convergence of the ensemble output to a Gaussian distribution only holds in the infinite width limit. There has been substantial work on finite-width corrections to the NTK limit (Huang & Yau, 2020; Yaida, 2020; Halverson et al., 2021; Erbin et al., 2022) which could in principle be used to quantify the resulting violations of exact equivariance. This is however of significant technical difficulty and therefore beyond the scope of this work. In the experimental section, we nevertheless demonstrate that even finite-width ensembles show emergent equivariance to good approximation.

## 7 EXPERIMENTS

In this section, we empirically study the emergent equivariance of finite width deep ensembles for several architectures (fully connected and convolutional), tasks (regression and classification), and application domains (computer vision and physics).

**Ising Model**  We validate our analytical computations with experiments on a problem for which we can compute the NTK exactly: the two-dimensional Ising model on a 5x5 lattice with energy function $\mathcal{E} = -J \sum_{\langle i,j \rangle} s_i s_j$, with the spins $s_i \in \{+1, -1\}$, $J$ a coupling constant and the sum runs over all adjacent spins. The energy of the Ising model is invariant under the cyclic group $C_4$ of rotations of the lattice by $90°$. We train ensembles of five different sizes with 100 to 10k members of fully-connected networks with hidden-layer widths 512, 1024 and 2048 to approximate the energy function using all rotations in $C_4$ as data augmentation. In this setting, we can compute the NTK exactly on the given training data using the JAX package `neural-tangents` (Novak et al., 2020). We verify that the ensembles converge to the NTK for large widths, see Appendix B.1.

To quantify the invariance of the ensembles, we measure the standard deviation of the predicted energy across the group orbit averaged over all datapoints of i) training set, ii) test set, and iii) out-of-distribution set. The latter is generated randomly drawing spins from a Gaussian distribution with mean zero and variance $400$. For better interpretability, we divide by the mean of $\mathcal{E}$, so that for a *relative standard deviation (RSD)* across orbits of one, the deviation from invariance is as large as a typical ground truth energy. For an exactly equivariant model, we would obtain an RSD of zero.

Figure 1 shows that the deep ensemble indeed exhibit the expected emergent invariance. As expected, the NTK features very low RSD compatible with numerical error. The RSD of the mean

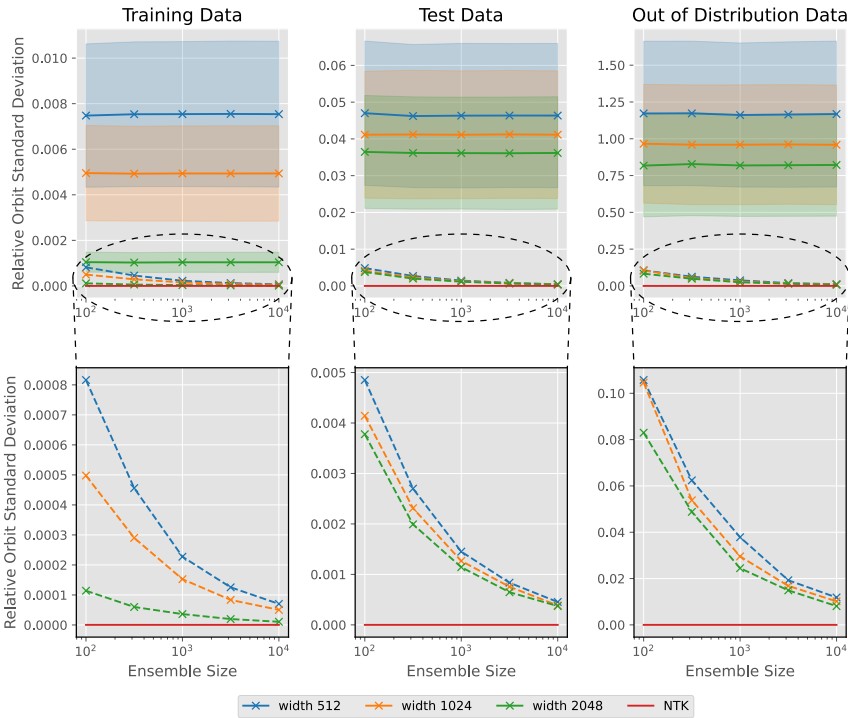

Figure 1: Invariance of predicted energies with respect to lattice rotations by $90°$. Solid lines refer to predictions of individual ensemble members and their standard deviation, dashed lines refer to mean predictions of the ensemble. Zoom-ins in the second row show that the invariance of mean predictions converges to NTK invariance for large ensembles and network widths.

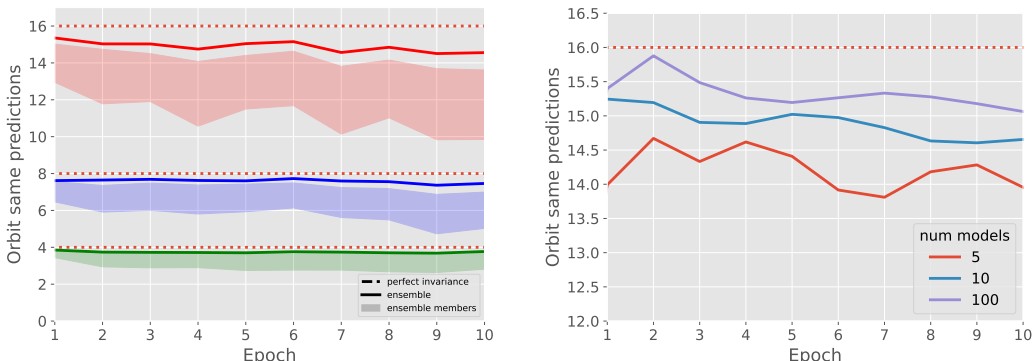

Figure 2: Emergent invariance for FashionMNIST **Left:** Number of out-of-distribution MNIST samples with the same prediction across a symmetry orbit for group orders 4 (green), 8 (blue), and 16 (red) versus training epoch. The models were trained on augmented FashionMNIST. Solid lines show the ensemble prediction. Shaded area is between the $25^{th}$ and $75^{th}$ quantile of the predictions of individual members of the ensemble. **Right:** Out of distribution invariance in the same setup as on the left-hand-side at group order 16. As the number of ensemble members increases, the prediction becomes more invariant, as expected.

predictions of the ensembles are larger but still very small and converge to the NTK results for large ensembles and network widths, cf. dashed lines in Figure 1. In contrast, the RSD computed for individual ensemble members is much higher and varies considerably between ensemble members, cf. solid lines in Figure 1. Even out of distribution, the ensemble means deviate from invariance only by about $0.8\%$ for large ensembles and network widths, compared to $82\%$ for individual ensemble members.

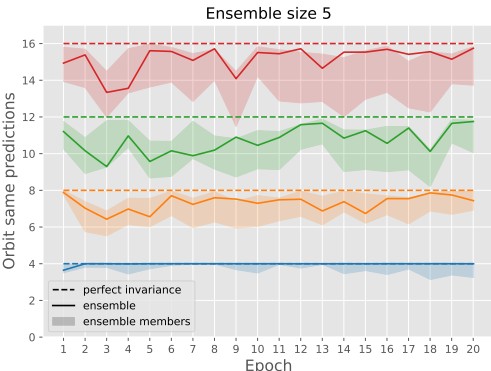 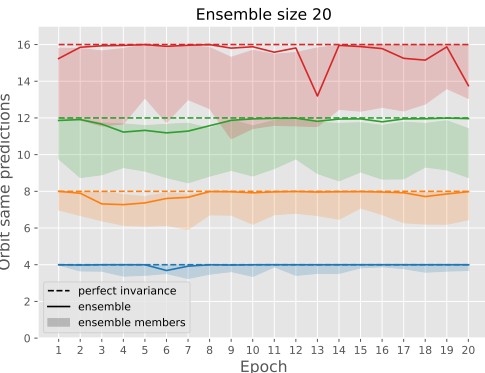

Figure 3: Ensemble invariance on OOD data for ensembles trained on histological data. Number of OOD samples with the same prediction across a symmetry orbit for group orders 4 (blue), 8 (orange), 12 (green) and 16 (red) versus training epoch. Even for ensemble size 5 (left), the ensemble predictions (solid line) are more invariant than the ensemble members (shaded region corresponding to 25th to 75th percentile of ensemble members). The effect is larger for ensemble size 20 (right).

**Rotated FashionMNIST**  We train convolutional neural networks on the an rotated FashionM-NIST dataset. Specifically, we augment the original dataset (Xiao et al., 2017) by all elements of the group orbit of the cyclic group $C_k$, i.e., all rotations of the image by any multiple of $360/k$ degrees with $k = 4, 8, 16$ and choose ensembles of size $M = 5, 10, 100$. We then evaluate the *orbit same prediction (OSP)*, i.e., how many of the images in a given group orbit have on average the same classification result as the unrotated image. We evaluate the OSP metric both on the validation set of FashionMNIST as well as on various out-of-distribution (OOD) datasets. Specifically, we choose the validation sets of MNIST, grey-scaled CIFAR10 as well as images for which each pixel is drawn iid from $\mathcal{N}(0, 1)$. Figure 2 shows the OSP metric for OOD data from MNIST. The ensemble prediction becomes more invariant as the number of ensemble members increases. Furthermore, the ensemble prediction is significantly more invariant as the individual ensemble members, i.e., the invariance is emergent. As the group order $k$ increases, more ensemble members are needed to achieve a high degree of invariance. More details about the experiments as well as plots showing results for the other OOD datasets can be found in Appendix B.2.

**Histological Data**  A realistic task, where rotational invariance is of key importance, is the classification of histological slices. We trained ensembles of CNNs on the NCT-CRC-HE-100K dataset (Kather et al., 2018) which comprises of stained histological images of human colorectal cancer and normal tissue with a resolution of $224 \times 224$ pixels in nine classes.

As for our experiments on FashionMNIST, we verify that the ensemble is more invariant as a function of its input than the ensemble members by evaluating the OSP on OOD data. In order to arrive at a sample of OOD data on which the network makes non-constant predictions, we optimize the input of the untrained ensemble to yield balanced predictions of high confidence. Using this specifically generated dataset for each ensemble, we observe the same increase in invariance also outside of the training domain as predicted by our theoretical considerations, cf. Figure 3. For further results on validation data as well as examples of our OOD data see Appendix D

## 8  CONCLUSIONS

Equivariant neural networks are a central ingredient in many machine learning setups, in particular in the natural sciences. However, constructing manifestly invariant models can be difficult. Deep ensembles are an important tool which can straightforwardly boost the performance and estimate uncertainty of existing models, explaining their widespread use in practice. In this work, using the theory of neural tangent kernels, we proved that infinitely wide ensembles show emergent equivariance when trained on augmented data. We furthermore discussed implications of finite width and ensemble size as well as the effect of approximating a continuous symmetry group. Experiments on several different datasets support our theoretical insights.

In future work, it would be interesting to incorporate the effects of finite width corrections and include a more detailed model of data augmentation, for instance along the lines of Dao et al. (2019).

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

## A  PROOFS

**Theorem 2** (Kernel invariance under orthogonal representations). *Let $G$ be a group and $\rho_X$ an orthogonal representation of $G$ acting on the input space $X$. Under the representation $\rho_X$, the neural tangent kernel $\Theta$ as defined in (3) as well as the NNGP kernel $\mathcal{K}$ as defined in (10) of a neural network satisfying the assumptions above are invariant:*

$$\Theta(x, x') = \Theta(\rho_X(g)x, \rho_X(g)x'),\tag{16}$$

$$\mathcal{K}(x, x') = \mathcal{K}(\rho_X(g)x, \rho_X(g)x').\tag{17}$$

*for all $g \in G$ and $x, x' \in X$.*

*Proof.* To simplify notation, we will use the shorthand notation

$$O = \rho_X(g)\tag{31}$$

for the orthogonal representation matrix corresponding to the action of group element $g$ on the input space $X$. This orthogonal matrix can be absorbed by redefining the parameters which multiply the input. For notational simplicity, we assume an MLP without biases in the following, but the proof immediately generalizes to the case of dependency on linear transformations of the input as stated in the assumptions.

Redefining the parameters $w^{(1)}$ of the first layer of the neural network $f_w : X \to Y$ yields

$$f_w(Ox) = f_{w'}(x),\tag{32}$$

where we have defined the new weights of layer $l$ as

$$w'^{(l)} = \begin{cases} w^{(1)}O, & \text{if } l = 1 \\ w^{(l)}, & \text{otherwise}. \end{cases}\tag{33}$$

We can use this result to rewrite the gradient of the network with respect to the parameters of the first layer

$$\frac{\partial}{\partial w_{ij}^{(1)}} f_w(Ox) = \frac{\partial}{\partial w_{ij}^{(1)}} f_{w'}(x) = \frac{\partial w'^{(1)}_{mn}}{\partial w_{ij}^{(1)}} \frac{\partial}{\partial w'^{(1)}_{mn}} f_{w'}(x) = O_{jn} \frac{\partial}{\partial w'^{(1)}_{in}} f_{w'}(x),\tag{34}$$

where here and in the following we use the Einstein summation convention. It is convenient to define

$$O^{(l)} = \begin{cases} O, & \text{if } l = 1 \\ \mathbb{I}, & \text{otherwise}, \end{cases}\tag{35}$$

such that the above gradient relation can be generalized to

$$\frac{\partial f_w(Ox)}{\partial w_{ij}^{(l)}} = O_{jn}^{(l)} \frac{\partial f_{w'}(x)}{\partial w'^{(l)}_{in}}.\tag{36}$$

Since the representation is orthogonal, it holds that for all $l \in \{1, \ldots, L\}$

$$(O^{(l)})^\top O^{(l)} = \mathbb{I} \iff (O^{(l)})_{jn} O_{jm}^{(l)} = \delta_{nm},\tag{37}$$

where $\delta_{nm}$ is the Kronecker symbol.

The neural tangent kernel (3) involves a sum over all layers and can thus be rewritten as

$$\Theta(Ox, Oy) = \sum_{l=1}^{L} \mathbb{E}_{w \sim p} \left[ \frac{\partial f_w(Ox)}{\partial w_{ij}^{(l)}} \frac{\partial f_w(Oy)}{\partial w_{ij}^{(l)}} \right]\tag{38}$$

$$= \sum_{l=1}^{L} \mathbb{E}_{w \sim p} \left[ \frac{\partial f_w(Ox)}{\partial w_{ij}^{(l)}} \frac{\partial f_w(Oy)}{\partial w_{ij}^{(l)}} \right]\tag{39}$$

$$= \sum_{l=1}^{L} \mathbb{E}_{w \sim p} \left[ \frac{\partial f_{w'}(x)}{\partial w'^{(l)}_{in}} \frac{\partial f_{w'}(y)}{\partial w'^{(l)}_{im}} (O^{(l)})_{jn} O_{jm}^{(l)} \right]\tag{40}$$

$$= \sum_{l=1}^{L} \mathbb{E}_{w \sim p} \left[ \frac{\partial f_{w'}(x)}{\partial w'^{(l)}_{in}} \frac{\partial f_{w'}(y)}{\partial w'^{(l)}_{in}} \right],\tag{41}$$

where we have used the relation between the original and redefined gradient (36) for the third equality and the unitarity of the representation (37) for the last, respectively.

The expectation value is over all weights and corresponds to the integral

$$\mathbb{E}_{w \sim p} = \prod_{k=1}^{L} \int \mathrm{d}w^{(k)} \, p(w^{(k)}) \,. \tag{42}$$

We now use the fact that a orthogonal transformation leaves the measure invariant

$$\mathrm{d}w^{(l)} p(w^{(l)}) = \left| \det \frac{\partial w^{(l)}}{\partial w'^l} \right| \mathrm{d}w'^{(l)} \, p(w'^{(l)}) = |\det O^\top| \, \mathrm{d}w'^{(l)} \, p(w'^{(l)}) = \mathrm{d}w'^{(l)} p(w'^{(l)}) \,. \tag{43}$$

Since a orthogonal matrix preserves the norm, i.e. $||w^{(l)}|| = ||O^{(l)} w^{(l)}|| = ||w'^{(l)}||$, the initialization density is invariant $p(w^{(l)}) = \frac{1}{Z} \exp(-\frac{||w^{(l)}||^2}{2\sigma_l^2}) = p(w'^{(l)})$. Using these results, we immediately conclude that

$$\Theta(Ox, Oy) = \sum_{l=1}^{L} \left( \prod_{k=1}^{L} \int \mathrm{d}w^{(k)} \, p(w^{(k)}) \right) \frac{\partial f_{w'}(x)}{\partial w_{in}'^{(l)}} \frac{\partial f_{w'}(y)}{\partial w_{in}'^{(l)}} \tag{44}$$

$$= \sum_{l=1}^{L} \left( \prod_{k=1}^{L} \int \mathrm{d}w'^{(k)} \, p(w'^{(k)}) \right) \frac{\partial f_{w'}(x)}{\partial w_{in}'^{(l)}} \frac{\partial f_{w'}(y)}{\partial w_{in}'^{(l)}} \tag{45}$$

$$= \sum_{l=1}^{L} \mathbb{E}_{w' \sim p} \left[ \frac{\partial f_{w'}(x)}{\partial w_{ij}'^{(l)}} \frac{\partial f_{w'}(y)}{\partial w_{ij}'^{(l)}} \right] \tag{46}$$

$$= \Theta(x, y) \tag{47}$$

This shows that the neural tangent kernel is invariant with respect to the orthogonal representation $\rho_X$.

The proof for the NNGP kernel is then completely analogous:

$$\mathcal{K}(Ox, Oy) = \mathbb{E}_{w \sim p} \left[ f_w(Ox) f_w(Oy) \right] = \mathbb{E}_{w' \sim p} \left[ f_{w'}(x) f_{w'}(y) \right] = \mathcal{K}(x, y) \,. \tag{48}$$

We stress that this proof critically relies on the structure of the neural tangent- and NNGP kernels. $\qquad\square$

**Lemma 7.** *Data augmentation implies that*

*(a)* $\Theta_{\pi_g(i), j} = \Theta_{i, \pi_g^{-1}(j)}$ *,*

*(b)* $\Theta_{\pi_g(i), j}^{-1} = \Theta_{i, \pi_g^{-1}(j)}^{-1}$ *,*

*(c)* $\mathcal{K}_{\pi_g(i), j} = \mathcal{K}_{i, \pi_g^{-1}(j)}$ *,*

*(d)* $\mathcal{K}_{\pi_g(i), j}^{-1} = \mathcal{K}_{i, \pi_g^{-1}(j)}^{-1}$ *,*

*and analogous results hold for any power of $\Theta$, $\Theta^{-1}$, $\mathcal{K}$ and $\mathcal{K}^{-1}$, respectively.*

*Proof.* **(a):** By data augmentation, it follows that

$$\begin{aligned}
\Theta_{\pi_g(i), j} &= \Theta(x_{\pi_g(i)}, x_j) \\
&= \Theta(\rho_X(g) x_i, x_j) \\
&= \Theta(x_i, \rho_X(g)^{-1} x_j) \\
&= \Theta(x_i, x_{\pi_g^{-1}(j)}) \\
&= \Theta_{i, \pi_g^{-1}(j)} \,.
\end{aligned} \tag{49}$$

For any power $N \in \mathbb{N}$ of the kernel, it holds therefore that

$$\begin{aligned}
\left[ \Theta^N \right]_{\pi_g(i), j} &= \Theta_{\pi_g(i), l} \left[ \Theta^{N-1} \right]_{lj} \\
&= \Theta_{i, \pi_g^{-1}(l)} \left[ \Theta^{N-1} \right]_{lj} \\
&\stackrel{l \mapsto \pi_g(l)}{=} \Theta_{il} \left[ \Theta^{N-1} \right]_{\pi_g(l)j} \\
&= \ldots = \left[ \Theta^{N-1} \right]_{il} \Theta_{l, \pi_g^{-1}(j)} \\
&= \left[ \Theta^N \right]_{i, \pi_g^{-1}(j)} \,.
\end{aligned}$$

**(b):** We start from the equality

$$\Theta(X, \rho_X(g)X)_{il} \left[\Theta(X, \rho_X(g)X)\right]_{lj}^{-1} = \delta_{ij} , \tag{50}$$

where we have used the following notation for the Gram matrix $\Theta(X, X)_{ij} := \Theta_{ij}$ and $G$ acts sample-wise on the dataset, $(\rho_X(g)X)_i = \rho_X(g)x_i$. By data augmentation, this can be rewritten as

$$\Theta(X, X)_{i, \pi_g(l)} \left[\Theta(X, \rho_X(g)X)\right]_{lj}^{-1} = \delta_{ij} . \tag{51}$$

We now relabel the summation variable $l \to \pi_g^{-1}(l)$ and obtain

$$\Theta(X, X)_{il} \left[\Theta(X, \rho_X(g)X)\right]_{\pi_g^{-1}(l), j}^{-1} = \delta_{ij} . \tag{52}$$

By uniqueness of the inverse matrix, it thus follows that

$$\Theta(X, X)_{lj}^{-1} = \left[\Theta(X, \rho_X(g)X)\right]_{\pi_g^{-1}(l), j}^{-1} \quad \Longleftrightarrow \quad \Theta(X, X)_{\pi_g(l), j}^{-1} = \left[\Theta(X, \rho_X(g)X)\right]_{lj}^{-1} . \tag{53}$$

Similarly, we can start from the expression

$$\left[\Theta(\rho_X(g)^{-1}X, X)\right]_{il}^{-1} \Theta(\rho_X^{-1}(g)X, X)_{lj} = \delta_{ij} . \tag{54}$$

By data augmentation, this can be rewritten as

$$\left[\Theta(\rho_X(g)^{-1}X, X)\right]_{il}^{-1} \Theta(X, X)_{\pi_g^{-1}(l), j} = \delta_{ij} . \tag{55}$$

Relabeling the summation variable $l \to \pi_g^{-1}(l)$, we obtain

$$\left[\Theta(\rho_X^{-1}(g)X, X)\right]_{i, \pi_g(l)}^{-1} \Theta(X, X)_{lj} = \delta_{ij} . \tag{56}$$

By uniqueness of the inverse matrix, it follows again that

$$\Theta(X, X)_{il}^{-1} = \left[\Theta(\rho_X^{-1}(g)X, X)\right]_{i, \pi_g(l)}^{-1} \quad \Longleftrightarrow \quad \Theta(X, X)_{i, \pi_g^{-1}(l)}^{-1} = \left[\Theta(\rho_X^{-1}(g)X, X)\right]_{il}^{-1} . \tag{57}$$

Combining the results (53) and (57), the statement of the theorem follows immediately:

$$\Theta_{\pi_g(i),j}^{-1} = \left[\Theta(X, X)\right]_{\pi_g(i),j}^{-1} \overset{(53)}{=} \left[\Theta(X, \rho_X(g)X)\right]_{ij}^{-1}$$
$$= \left[\Theta(\rho_X^{-1}(g)X, X)\right]_{ij}^{-1} \overset{(57)}{=} \Theta(X, X)_{i, \pi_g^{-1}(j)}^{-1} = \Theta_{i, \pi_g^{-1}(j)}^{-1} . \tag{58}$$

The proof for any power $(\Theta^{-1})^N$ of the inverse Gram matrix follows in complete analogy to the proof of the same result for the Gram matrix $\Theta$.

**(c):** The proof for the NNGP follows in close analogy to the one for the NTK, see (a):

$$\mathcal{K}_{\pi_g(i),j} = \mathcal{K}(x_{\pi_g(i)}, x_j) = \mathcal{K}(\rho_X(g)x_i, x_j) = \mathcal{K}(x_i, \rho_X^{-1}(g)x_j) = \mathcal{K}(x_i, x_{\pi_g^{-1}(j)}) = \mathcal{K}_{i, \pi_g^{-1}(j)} . \tag{59}$$

The proof for any power of the NNGP again follows in complete analogy to (a).

**(d):** Since the transformation properties of $\Theta$ and $\mathcal{K}$ under $G$ are completely identical, the proof follows the steps of (b) verbatim with the replacement $\Theta \to \mathcal{K}$. Similarly for any power of $\mathcal{K}$. $\qquad \square$

Using this result, we can then show the following lemma as stated in the main part:

**Lemma 3** (Shift of permutation). *Data augmentation implies that one can shift the permutation group action from the first index to the second index for any matrix-valued analytical function $F$ involving the Gram matrices of the NNGP and NTK as well as their inverses:*

$$F(\Theta, \Theta^{-1}, \mathcal{K}, \mathcal{K}^{-1})_{\pi_g(i),j} = F(\Theta, \Theta^{-1}, \mathcal{K}, \mathcal{K}^{-1})_{i, \pi_g^{-1}(j)} . \tag{20}$$

*where $\pi_g$ denotes the group action in terms of training set permutations, see (13).*

*Proof.* As the matrix-valued function $F$ is analytic, it has the following series expansion

$$F(\Theta, \Theta^{-1}, \mathcal{K}, \mathcal{K}^{-1})_{ij} = \sum_{n=1}^{\infty} \sum_{P_n} c_{P_n} P_n(\Theta, \Theta^{-1}, \mathcal{K}, \mathcal{K}^{-1})_{ij} , \tag{60}$$

where the inner sum is over all order $n$ polynomials involving $\Theta$ and $\mathcal{K}$ as well as their inverses and $c_{P_n}$ are coefficients.

By Lemma 7, for any such polynomial $P_n$ it holds that

$$P_n(\Theta, \Theta^{-1}, \mathcal{K}, \mathcal{K}^{-1})_{\pi_g(i)j} = P_n(\Theta, \Theta^{-1}, \mathcal{K}, \mathcal{K}^{-1})_{i\pi_g^{-1}(j)} . \tag{61}$$

Applying this result to the series expansion above immediately implies the claim of the lemma. $\qquad \square$

**Theorem 4** (Emergent Equivariance of Deep Ensembles). *The distribution of (possibly vector-valued) large-width ensemble members $f_w : X \to Y$ is equivariant with respect to the orthogonal representations $\rho_X$ and $\rho_Y$ of the group $G$ if data augmentation is applied. In particular, the ensemble is equivariant,*

$$\bar{f}_t(\rho_X(g)\,x) = \rho_Y(g)\,\bar{f}_t(x) \tag{27}$$

*for all $g \in G$. This result holds*

1. *at any training time $t$,*

2. *for any element of the input space $x \in X$.*

*Proof.* In the case of vector-valued networks, the definition (3) has to be extended to

$$\boldsymbol{\Theta}^{\alpha\beta}(x, x') = \sum_{l=1}^{L} \mathbb{E}_{w \sim p} \left[ \left( \frac{\partial f^\alpha(x)}{\partial w^{(l)}} \right)^\top \frac{\partial f_w^\beta(x')}{\partial w^{(l)}} \right], \tag{62}$$

where $\alpha, \beta$ are component indices for the output vector. It can be shown (Lee et al., 2019) that in the infinite width limit, the NTK is proportional to the unit matrix with respect to the output indices

$$\boldsymbol{\Theta}^{\alpha\beta}(x, x') = \delta^{\alpha\beta}\Theta(x, x') \qquad \text{with} \qquad \Theta(x, x') = \sum_{l=1}^{L} \mathbb{E}_{w \sim p} \left[ \left( \frac{\partial f_w^\gamma(x)}{\partial w^{(l)}} \right)^\top \frac{\partial f_w^\gamma(x')}{\partial w^{(l)}} \right], \tag{63}$$

where $\delta$ is the Kronecker symbol and $\Theta(x, x')$ does not depend on the index $\gamma$ in the expectation value. For the vector-valued case, the mean prediction of the ensemble is given by

$$\mu_t^\alpha(x) = \boldsymbol{\Theta}^{\alpha\beta}(x, x_i) \left[ \boldsymbol{\Theta}^{-1} \left( \mathbb{I} - \exp(-\eta\boldsymbol{\Theta}t) \right) \right]_{ij}^{\beta\gamma} y_j^\gamma \tag{64}$$

$$= \Theta(x, x_i) \left[ \Theta^{-1} \left( \mathbb{I} - \exp(-\eta\Theta t) \right) \right]_{ij} y_j^\alpha, \tag{65}$$

where we use the Einstein summation convention to sum over output-component indices as well. To show that $\mu_t^\alpha(x)$ is equivariant, we consider the mean at a transformed test sample $\rho_X(g)x$

$$\mu_t^\alpha(\rho_X(g)x) = \Theta(\rho_X(g)x, x_i)\,\Theta_{ij}^{-1}\left( \mathbb{I} - \exp(-\eta\Theta t) \right)_{jk} y_k^\alpha \tag{66}$$

$$= \Theta(x, \rho_X^{-1}(g)x_i)\,\Theta_{ij}^{-1}\left( \mathbb{I} - \exp(-\eta\Theta t) \right)_{jk} y_k^\alpha \tag{67}$$

$$= \Theta(x, x_{\pi_g^{-1}(i)})\,\Theta_{ij}^{-1}\left( \mathbb{I} - \exp(-\eta\Theta t) \right)_{jk} y_k^\alpha, \tag{68}$$

where we have used that Theorem 2 implies that $\Theta(\rho_X(g)x, x_i) = \Theta(\rho_X^{-1}(g)\rho_X(g)x, \rho_X^{-1}(g)x_i) = \Theta(x, \rho_X^{-1}(g)x_i)$ for the second equality and used data augmentation in the last step. We now redefine the summation variable $i \to \pi_g(i)$ and thus obtain

$$\mu_t^\alpha(\rho_X(g)x) = \Theta(x, x_i)\,\Theta_{\pi_g(i)j}^{-1}\left( \mathbb{I} - \exp(-\eta\Theta t) \right)_{jk} y_k^\alpha, \tag{69}$$

By Lemma 3, we can rewrite this as

$$\mu_t^\alpha(\rho_X(g)x) = \Theta(x, x_i)\,\Theta_{ij}^{-1}\left( \mathbb{I} - \exp(-\eta\Theta t) \right)_{j,\pi_g^{-1}(k)} y_k^\alpha, \tag{70}$$

We now again redefine the summation variable $k \to \pi_g(k)$ and use the data augmentation property (13) of the labels $y$, i.e. $y_{\pi_g(k)} = \rho_Y(g)y_k$, to obtain

$$\mu_t^\alpha(\rho_X(g)x) = \Theta(x, x_i)\,\Theta_{ij}^{-1}\left( \mathbb{I} - \exp(-\eta\Theta t) \right)_{j,k} y_{\pi_g(k)}^\alpha \tag{71}$$

$$= \Theta(x, x_i)\,\Theta_{ij}^{-1}\left( \mathbb{I} - \exp(-\eta\Theta t) \right)_{j,k} \rho_Y^{\alpha\beta}(g)y_k^\beta \tag{72}$$

$$= (\rho_Y(g)\mu_t(x))^\alpha. \tag{73}$$

This establishes that the mean (and therefore the ensemble) is equivariant.

The proof for the variance follows very similar argument. In particular, as the NTK, the NNGP becomes diagonal in the infinite width limit as well (Lee et al., 2019)

$$\mathcal{K}^{\alpha\beta}(x, x') = \mathbb{E}_{w \sim p} \left[ f_w^\alpha(x)\, f_w^\beta(x') \right] = \delta^{\alpha\beta} \mathbb{E}_{w \sim p} \left[ f_w^\gamma(x)\, f_w^\gamma(x') \right] = \delta^{\alpha\beta} \mathcal{K}(x, x'). \tag{74}$$

Consequently, also the covariance $\Sigma$ is proportional to the unit matrix with respect to the output indices

$$\boldsymbol{\Sigma}^{\alpha\beta}(x, x') = \delta^{\alpha\beta}\Sigma(x, x'), \tag{75}$$

with

$$\Sigma_t(x,x') = \mathcal{K}(x,x') + \Sigma_t^{(1)}(x,x') - (\Sigma_t^{(2)}(x,x') + \text{h.c.}) . \tag{76}$$

We will now show that $\Sigma(x,x')$ is invariant under $G$ by considering each summand individually. For the first summand, invariance $\mathcal{K}(\rho_X(g)x, \rho_X(g)x') = \mathcal{K}(x,x')$ is an immediate consequence of Lemma 2. Invariance of the second summand follows by

$$\Sigma_t^{(1)}(\rho_X(g)x, \rho_X(g)x') = \Theta(\rho_X(g)x, x_i)\, \Theta_{ij}^{-1}\, (\mathbb{I}-e^{-\eta\Theta t})_{jk}\, \mathcal{K}_{kl}\, (\mathbb{I}-e^{-\eta\Theta t})_{ls}\, \Theta_{sh}^{-1}\, \Theta(x_h, \rho_X(g)x') \tag{77}$$

$$= \Theta(x, \rho_X^{-1}(g)x_i)\, \Theta_{ij}^{-1}\, (\mathbb{I}-e^{-\eta\Theta t})_{jk}\, \mathcal{K}_{kl}\, (\mathbb{I}-e^{-\eta\Theta t})_{ls}\, \Theta_{sh}^{-1}\, \Theta(x_h, \rho_X(g)x') \tag{78}$$

$$= \Theta(x, x_{\pi_g^{-1}(i)})\, \Theta_{ij}^{-1}\, (\mathbb{I}-e^{-\eta\Theta t})_{jk}\, \mathcal{K}_{kl}\, (\mathbb{I}-e^{-\eta\Theta t})_{ls}\, \Theta_{sh}^{-1}\, \Theta(x_h, \rho_X(g)x') \tag{79}$$

$$= \Theta(x, x_i)\, \Theta_{\pi_g(i),\,j}^{-1}\, (\mathbb{I}-e^{-\eta\Theta t})_{jk}\, \mathcal{K}_{kl}\, (\mathbb{I}-e^{-\eta\Theta t})_{ls}\, \Theta_{sh}^{-1}\, \Theta(x_h, \rho_X(g)x') \tag{80}$$

$$= \Theta(x, x_i)\, \Theta_{ij}^{-1}\, (\mathbb{I}-e^{-\eta\Theta t})_{jk}\, \mathcal{K}_{kl}\, (\mathbb{I}-e^{-\eta\Theta t})_{ls}\, \Theta_{sh}^{-1}\, \Theta(x_{\pi_g(h)}, \rho_X(g)x') \tag{81}$$

$$= \Theta(x, x_i)\, \Theta_{ij}^{-1}\, (\mathbb{I}-e^{-\eta\Theta t})_{jk}\, \mathcal{K}_{kl}\, (\mathbb{I}-e^{-\eta\Theta t})_{ls}\, \Theta_{sh}^{-1}\, \Theta(\rho_X(g)x_h, \rho_X(g)x') \tag{82}$$

$$= \Theta(x, x_i)\, \Theta_{ij}^{-1}\, (\mathbb{I}-e^{-\eta\Theta t})_{jk}\, \mathcal{K}_{kl}\, (\mathbb{I}-e^{-\eta\Theta t})_{ls}\, \Theta_{sh}^{-1}\, \Theta(x_h, x') \tag{83}$$

$$= \Sigma_t^{(1)}(x, x') . \tag{84}$$

The invariance $\Sigma_t^{(2)}(\rho_X(g)x, \rho_X(g)x') = \Sigma_t^{(2)}(x,x')$ follows completely analogously.

Since ensemble members follow a Gaussian process with mean function $\mu_t(x)$ and covariance function $\Sigma_t(x,x')$ which is proportional to the unit matrix with respect to output indices, the equivariance of $\mu_t(x)$ and invariance of $\Sigma_t(x,x')$ together imply that the distribution of the ensemble members is equivariant.

$\square$

## A.1 FINITE NUMBER OF ENSEMBLE MEMBERS

**Lemma 8.** *The probability that the deep ensemble $\bar{f}_t$ and its estimate $\hat{f}_t$ differ by more than a given threshold $\delta$ is bounded by*

$$\mathbb{P}\left[|\hat{f}_t(x) - \bar{f}_t(x)| > \delta\right] \leq \sqrt{\frac{2}{\pi}} \frac{\sigma_x}{\delta} \exp\left(-\frac{\delta^2}{2\sigma_x^2}\right) , \tag{85}$$

*where we have defined*

$$\sigma_x^2 := Var(\hat{f}_t)(x) = \frac{\Sigma_t(x)}{M} \tag{86}$$

*with the output variance $\Sigma_t(x) = \Sigma_t(x,x)$ defined in (7).*

*Proof.* The probability of such deviations is given by

$$\mathbb{P}\left[|\hat{f}_t(x) - \bar{f}_t(x)| > \delta\right] = \frac{2}{\sqrt{2\pi}\sigma_x} \int_\delta^\infty \exp\left(-\frac{t^2}{2\sigma_x^2}\right) dt \tag{87}$$

We now change the integration variable to $\tau = \frac{t}{\sigma_x \sqrt{2}}$ and obtain

$$\mathbb{P}\left[|\hat{f}_t(x) - \bar{f}(_t x)| > \delta\right] = \frac{2}{\sqrt{\pi}} \int_{\frac{\delta}{\sqrt{2}\sigma_x}}^\infty \exp(-\tau^2) d\tau \leq \frac{1}{\sqrt{\pi}} \frac{\sqrt{2}\sigma_x}{\delta} \int_{\frac{\delta}{\sqrt{2}\sigma_x}}^\infty (2\tau)\exp(-\tau^2) d\tau , \tag{88}$$

where we have used that $1 \leq \frac{2\tau}{2\min(\tau)}$ for $\tau \geq \min(\tau)$ to obtain the last inequality. The integral can be straightforwardly evaluated by rewriting the integrand as a total derivative and we thus obtain

$$\mathbb{P}\left[|\hat{f}_t(x) - \bar{f}_t(x)| > \delta\right] \leq \sqrt{\frac{2}{\pi}} \frac{\sigma_x}{\delta} \exp\left(-\frac{\delta^2}{2\sigma_x^2}\right) . \tag{89}$$

$\square$

We stress that this result holds for any Monte-Carlo estimator and we therefore suspect that it could be well-known. For most MC estimators, it is however of relatively little use as the variance $\Sigma$ is not known in closed form — in stark contrast to the deep ensemble, see (7), considered in this paper. This could explain why we were not able to locate this result in the literature.

For the deep ensemble, we can therefore exactly determine the necessary number of ensemble size to stay within a certain threshold $\delta$ with a given probability $1 - \epsilon$. For this, one has to set the right-hand-side of the derived expression to this confidence $\epsilon$ and solve for the necessary ensemble size $M$. However, this equation appears to have no closed-from solution and needs to be solved numerically. We advise the reader to do so if need for a tight bound arises. For the presentation in the main part, we however wanted to derive a closed-form solution for $M$ and thus had to rely on a looser bound which implies the following statement:

**Lemma 5** (Bound for finite ensemble members). *The deep ensemble $\bar{f}_t$ and its estimate $\hat{f}_t$ do not differ by more than threshold $\delta$,*

$$|\bar{f}_t(x) - \hat{f}_t(x)| < \delta \,, \tag{28}$$

*with probability $1 - \epsilon$ for ensemble sizes $M$ that obey*

$$M > -\frac{2\Sigma_t(x)}{\delta^2} \ln\left(\sqrt{\pi}\epsilon\right) \,. \tag{29}$$

*Proof.*

$$\mathbb{P}\left[|\hat{f}_t(x) - \bar{f}_t(x)| > \delta\right] < \frac{1}{\sqrt{\pi}} \frac{1}{z} \exp\left(-z^2\right) \leq \frac{1}{\sqrt{\pi}} \exp\left(-z^2\right) \overset{!}{<} \epsilon \tag{90}$$

with $z = \frac{\delta}{\sqrt{2}\sigma_x}$ and where we assume that $M$ is chosen sufficiently large such that $z \geq 1$. This implies that

$$z^2 > -\ln(\sqrt{\pi}\epsilon) \qquad \Leftrightarrow \qquad M > -\frac{2\Sigma_t(x)}{\delta^2} \ln(\sqrt{\pi}\epsilon) \,. \tag{91}$$

$\square$

## A.2 Continuous Groups

**Lemma 6** (Bound for continuous groups). *Consider a deep ensemble of neural networks with Lipschitz continuous derivatives with respect to the parameters. For an approximation $A \subset G$ of a continuous symmetry group G with discretization error $\epsilon$, the prediction of the ensemble trained on A deviates from invariance by*

$$|\bar{f}_t(x) - \bar{f}_t(\rho_X(g)\,x)| \leq \epsilon\,C(x)\,, \qquad \forall g \in G\,,$$

*where C is independent of g.*

*Proof.* As described in the main text, we consider a finite subgroup $A \subset G$ which we use for data augmentation (instead of using the continuous group $G$). The discretization error for the representation $\rho_X$ is given by

$$\epsilon = \max_{g \in G} \min_{g' \in A} ||\rho_X(g) - \rho_X(g')|| \,. \tag{92}$$

This implies that for any $g \in G$, we can find a $g' \in A$ such that

$$||\rho_X(g)x_i - x_{\pi_{g'}(i)}|| = ||\rho_X(g)x_i - \rho_X(g')x_i|| \leq ||\rho_X(g) - \rho_X(g')||\,||x_i|| < \epsilon||x_i||\,, \tag{93}$$

where we have used data augmentation (13) over $A$.

We can then calculate the difference of the prediction at any test point $x$ and its transformation:

$$|\bar{f}_t(x) - \bar{f}_t(\rho_X(g)x)| = |\mu_t(x) - \mu_t(\rho_X(g)x)| \tag{94}$$

$$= |(\Theta(x, x_i) - \Theta(\rho_X(g)x, x_i))\,\Theta_{ij}^{-1}\,(\mathbb{I} - \exp(-\eta\Theta t))_{jk}\,y_k| \tag{95}$$

From the Lemma 3, it follows that

$$\Theta(x, x_i)\,\Theta_{ij}^{-1}\,(\mathbb{I} - \exp(-\eta\Theta t))_{jk}\,y_k = \Theta(x, x_i)\,\Theta_{ij}^{-1}\,(\mathbb{I} - \exp(-\eta\Theta t))_{jk}\,y_{\pi_{g'}(k)} \tag{96}$$

$$= \Theta(x, x_{\pi_{g'}^{-1}(i)})\,\Theta_{ij}^{-1}\,(\mathbb{I} - \exp(-\eta\Theta t))_{jk}\,y_k \tag{97}$$

Thus the difference can be rewritten as follows

$$|\bar{f}_t(x) - \bar{f}_t(\rho_X(g)x)| = |(\Theta(x, x_{\pi_{g'}^{-1}(i)}) - \Theta(\rho_X(g)x, x_i))\,\Theta_{ij}^{-1}\,(\mathbb{I} - \exp(-\eta\Theta t))_{jk}\,y_k| \tag{98}$$

$$= |(\Theta(x, x_{\pi_{g'}^{-1}(i)}) - \Theta(x, \rho_X^{-1}(g)x_i))\,\Theta_{ij}^{-1}\,(\mathbb{I} - \exp(-\eta\Theta t))_{jk}\,y_k| \tag{99}$$

It is convenient to define

$$\Delta\Theta(x', x, \bar{x}) \equiv |\Theta(x', x) - \Theta(x', \bar{x})| \tag{100}$$

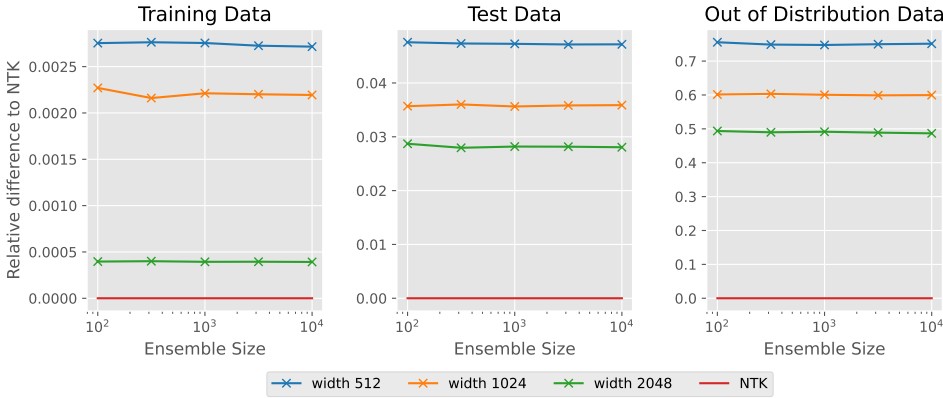

Figure 4: Difference in relative predicted total energy $\mathcal{E}$ between the ensembles and the NTK on the training data, in-distribution test data and out of distribution.

which can be bounded as follows

$$\Delta\Theta(x', x, \bar{x}) = \left| \sum_{l=1}^{L} \mathbb{E}_{w \sim p} \left[ \left( \frac{\partial f_w(x')}{\partial w^{(l)}} \right)^{\top} \left( \frac{\partial f_w(x)}{\partial w^{(l)}} - \frac{\partial f_w(\bar{x})}{\partial w^{(l)}} \right) \right] \right| \tag{101}$$

$$\leq ||x - \bar{x}|| \sum_{l=1}^{L} \mathbb{E}_{w \sim p} \left[ \left\| \left( \frac{\partial f_w(x')}{\partial w^{(l)}} \right)^{\top} \cdot L(w^{(l)}) \right\| \right] \tag{102}$$

$$\equiv ||x - \bar{x}|| \, \hat{C}(x) , \tag{103}$$

where $L(w^{(l)})$ is the Lipschitz constant of $\partial_{w^{(l)}} f_w$ and we emphasize that the norm is with respect to the input space. Using this expression, we can bound the difference of the means (99) by using the triangle inequality

$$|\bar{f}_t(x) - \bar{f}_t(\rho_X(g)x)| \leq \hat{C}(x) \sqrt{\sum_i ||x_{\pi_{g'}^{-1}(i)} - \rho_X(g)^{-1}x_i||^2} \sqrt{\sum_i (\sum_{j,k} \Theta_{ij}^{-1} \, (\mathbb{I} - \exp(-\eta\Theta t))_{jk} \, y_k])^2}$$

$$\leq \epsilon \, \hat{C}(x) \sqrt{\sum_i ||x_i||^2} \sqrt{\sum_i (\sum_{j,k} \Theta_{ij}^{-1} \, (\mathbb{I} - \exp(-\eta\Theta t))_{jk} \, y_k])^2} \equiv \epsilon C(x) .$$

Note that this result suggests that one should choose the discretization carefully to achieve as tight of a bound as possible. □

## B EXPERIMENTS

In this section, we provide further details about our experiments.

### B.1 ISING MODEL

**Training details** The energy function of the Ising model can be written as

$$\mathcal{E} = -\frac{J}{\text{vol}(L)} \sum_{i \in L} E(i) , \tag{104}$$

where $J$ is a coupling constant which we set to one for convenience and $\text{vol}(L)$ denotes the number of lattice sites. The local energy $E(i)$ is given by[1]

$$E(i) = \sum_{j \in \mathcal{N}(i)} s_i s_j , \tag{105}$$

where $\mathcal{N}(i)$ denotes the neighbors of $i$ along the lattice axes. The expectation value of $\mathcal{E}$ vanishes and its standard deviation is 2 for uniform sampling of spins in $\{+1, -1\}$.

---

[1] Usually, one only sums over pairs of spins. Our prescription differs from that convention by an irrelevant factor of two and makes the local energy exactly equivariant under rotations of the lattice by $90°$.

The energy of the Ising model is invariant under rotations of the lattice by $90°$, since the local energy (105) stays invariant if the neighborhood is rotated and the sum in (104) is just reshuffled. We train a fully-connected network with one hidden layer and a ReLU activation on 128 samples augmented with full $C_4$ orbits to 512 training samples. To obtain a sufficient training signal, we train the networks with a squared error loss on the local energies (105). We train for 100k steps of full-batch gradient descent with learning rate 0.5 for network widths 128, 512 and 1024 and learning rate 1.0 for network width 2048.

**Ensemble-convergence to the NTK**  We verify that the ensembles converge to the NTK for large widths by computing the difference in total energy $\mathcal{E}$ between the mean ensemble prediction and the predicted mean of the NTK, cf. Figure 4. To make the numbers easily interpretable, we plot the relative difference, where we divide by the standard deviation of the ground truth energy, 2, which gives a typical value for $\mathcal{E}$. We perform the comparisons on the training data, in-distribution test data and out of distribution data. As expected, agreement is highest on the training data and lowest out of distribution, but in each case, ensembles with higher-width hidden layer generate mean predictions closer to the NTK. Beyond ensemble size 1000, the estimate of the expectation value over initializations in the NTK seems to be accurate enough that no further fluctuations can be seen in the plots.

### B.2  ROTATED FASHIONMNIST

**Ensemble architecture**  As ensemble members, we use a simple convolutional neural network with two convolutional layers of kernel size 3 and 6 as well as 16 channels respectively. Both convolutional layers are followed by a relu non-linearity as well as $2 \times 2$ max-pooling. This is then followed by layers fully-connected of size $(400, 120)$, $(120, 84)$, and $(84, 10)$ of which the first two are fed into relu non-linearities. We choose ensembles of size $M = 5, 10, 100$.

**OOD data**  We use the validation set of greyscaled and rescaled CIFAR10, the validation set of MNIST, as well as a dataset generated by images with pixels drawn iid from $N(0, 1)$ as OOD data. We also evaluate the invariance on the validation set of FMNIST, i.e., on in-distribution data. Please refer to the corresponding Figure 7, 8, and 9 contained in this appendix for the results.

**Data augmentation**  We augment the original dataset by all elements of the group orbit of the cyclic group $C_k$, i.e., all rotations of the image by any multiple of $360/k$ degrees and ensure that each epoch contains all element of the group orbit in each epoch to closely align the experiments with our theoretical analysis. However, in exploratory analysis, we did not observe a notable difference when applying random group elements in each training step. For the cyclic group $C_k$, we choose group orders $k = 4, 8, 16$.

**Training details**  We use the ADAM optimizer with the standard learning rate of pytorch lightning, i.e., 1e-3. We train for 10 epochs on the augmented dataset. We evaluate the metrics after each epoch on both the in-distribution and the out-of-distribution data. The ensembles achieve a test accuracy on the augmented datasets of between 88 to 91 percent depending on the chosen group order and ensemble size.

**OSP metric:**  To obtain the orbit same prediction, we measure

$$\sum_{g \in G} \mathbb{I}(\mathrm{argmax}_\alpha f^\alpha(\rho_X(g)x), \mathrm{argmax}_\alpha f^\alpha(x)),  \tag{106}$$

where $\mathbb{I}$ denotes the indicator function. This corresponds to the number of elements in the orbit that have the same predicted class as the transformed data sample $x$. The orbit same prediction (OSP) of a dataset $\mathcal{D}$ is then this number averaged over all elements in the dataset. Note that the OSP has minimal value 1 as the identity is always part of the orbit.

**Continuous rotations:**  We analyze the generalization properties to the full two-dimensional rotation group $SO(2)$ for deep ensembles trained with data augmentation using the finite cyclic group $C_k$. To this end, we define the continuous orbit same prediction as:

$$\frac{1}{\mathrm{Vol}(SO(2))} \int_{SO(2)} \mathrm{d}g\, \mathbb{I}(\mathrm{argmax}_\alpha f^\alpha(\rho_X(g)x), \mathrm{argmax}_\alpha f^\alpha(x)),  \tag{107}$$

where $\mathrm{d}g$ denotes the Haar measure. This continuous orbit same prediction thus corresponds to the percentage of elements in the orbit that are classified the same way as the untransformed element. We estimate this quantity by Monte-Carlo. The results of our analysis are shown in Figure 5 and clearly establish that for sufficiently high group order of the cyclic group used for data augmentation, the ensemble is approximately invariant with respect to the continuous symmetry as well. In particular, it is signficantly more invariant as its ensemble members. Interestingly, this is competitive with a model that is using canonicalization Kaba et al. (2023) with

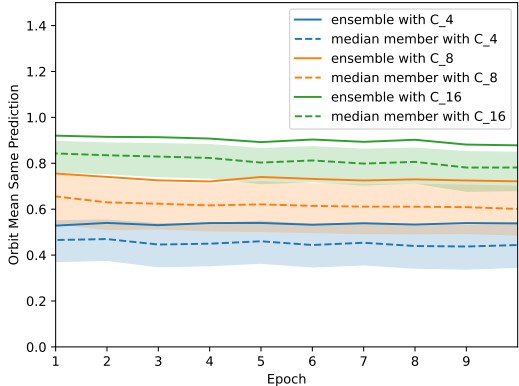

Figure 5: Mean orbit same prediction over $SO(2)$ group orbits. Solid lines show the ensemble prediction while dotted lines show the median of the ensemble members. Error band denotes the 75th and 25th percentile. As the group order $k$ of the cyclic group $C_k$ used for data augmentation increases, the mean orbit same prediction over $SO(2)$ increases. For $k = 16$, over 90 percent of the orbit elements have the same prediction as the untransformed input establishing that the model is approximately invariant under the continuous symmetry as well. The invariance of the ensemble is again emergent in the sense that it is above the 75th percentile of the ensemble members.

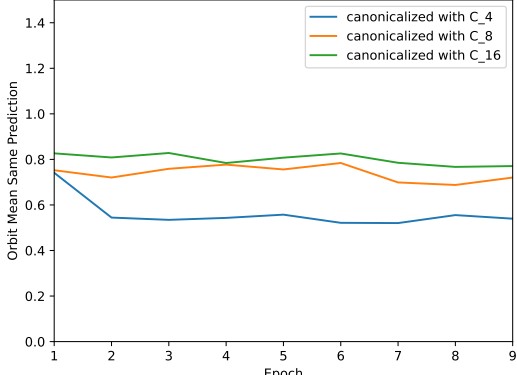

Figure 6: Mean orbit same prediction over $SO(2)$ group orbits for a model canonicalized with respect to $C_k$. As the group order $k$ of the cyclic group $C_k$ used for data augmentation increases, the mean orbit same prediction over $SO(2)$ increases.

respect to $C_k$ and the same network architecture as its predictor network. This comes however with important caveats: canonicalization can be performed with respect to the full $SO(2)$ equivariance. Furthermore, we compare a single canonicalized model to an ensemble. For this reason, we stress that we do not want to claim any preference of deep ensembles over canonicalization. Rather, we believe that emergent equivariance should be used in situations for which a deep ensemble is also required for other reasons, such as uncertainty prediction.

## C   CROSS PRODUCT

**Training**   We train ensembles of two hidden-layer fully-connected networks to predict the cross-product $x \times y$ in $\mathbb{R}^3$ given two vectors $x$ and $y$. This task is equivariant with respect to rotations $R \in \mathrm{SO}(3)$,

$$Rx \times Ry = R(x \times y).\tag{108}$$

The training data consists of 100 vector pairs with components sampled from $\mathcal{N}(0, 1)$, the validation data consists of 1000 such pairs. For out of distribution data, we sample from $\mathcal{N}(0, 100)$. We train using 10-fold data augmentation, i.e. we sample 10 rotation matrices from SO(3) and rotate the training data with these

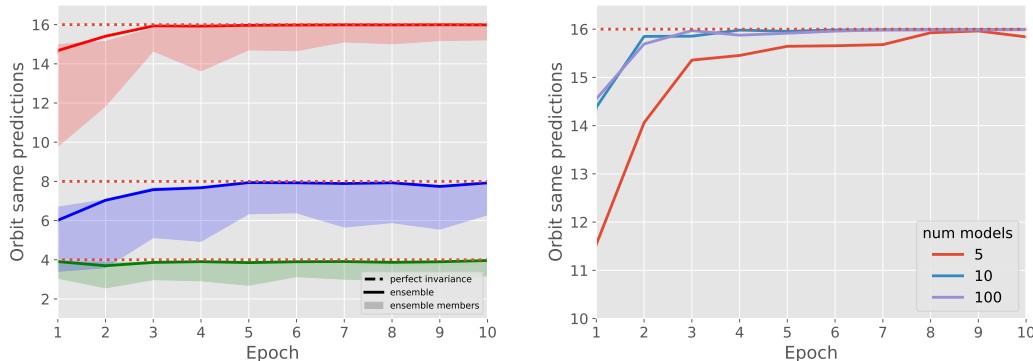

Figure 7: Same as Figure 2 but for OOD images with pixels drawn iid from $N(0, 1)$.

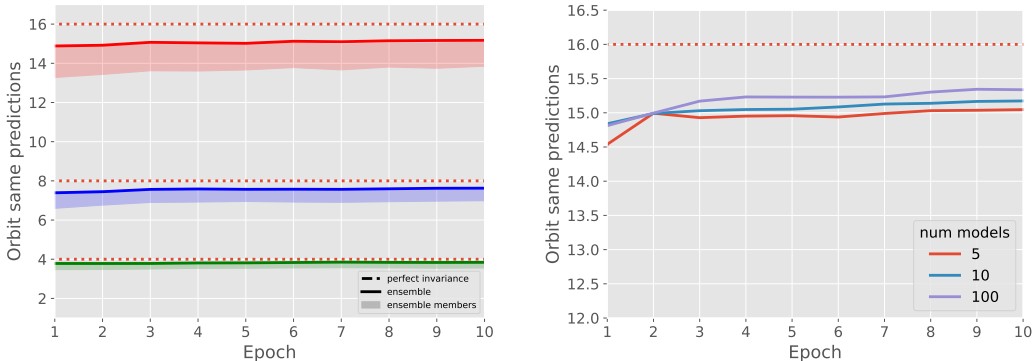

Figure 8: Same as Figure 2 but for FMNIST, i.e., in-distribution data.

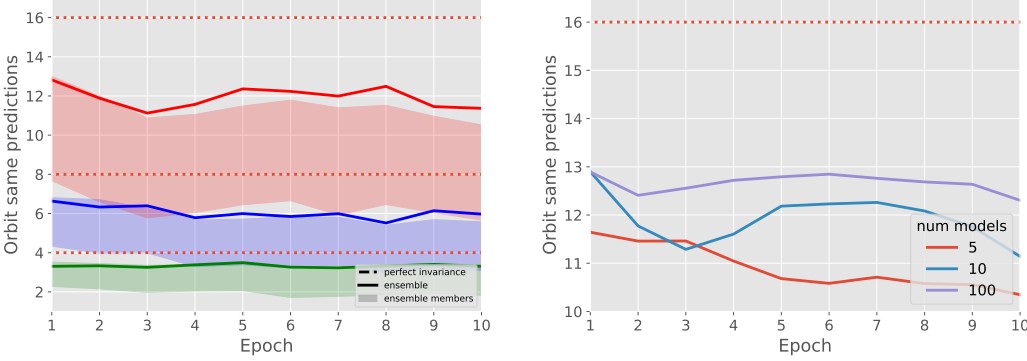

Figure 9: Same as Figure 2 but for rescaled and greyscaled CIFAR10 OOD data.

matrices, resulting in 1000 training vector pairs. We train for 50 epochs using the Adam optimizer and reach validation RMSEs of about 0.3 with exact performance depending on layer width and ensemble size.

**Orbit MSE**   To evaluate how equivariant the ensembles trained with data augmentation are on a given dataset, we sample 100 rotation matrices from SO(3) and augment each input vector pair with their 100 rotated versions. Then, we predict the cross products on this enlarged dataset and rotate the predicted vectors back using the inverse rotations. Finally, we measure the MSE across the 100 back-rotated predictions against the unrotated prediction. The orbit MSE is averaged over the last five epochs.

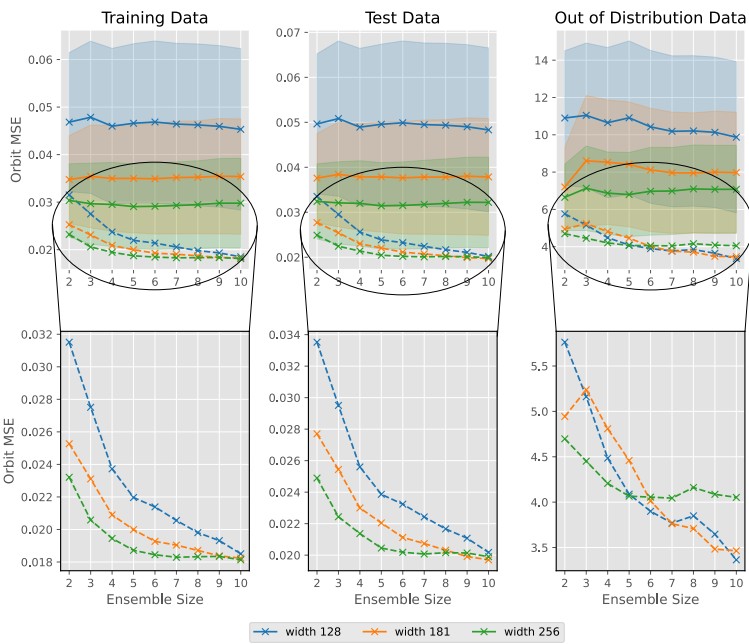

Figure 10: Emerging equivariance of ensembles predicting the cross-product. Plotted is the MSE of predictions across a random 100-element subset of the symmetry orbit of SO(3) versus ensemble size. Solid lines refer to the orbit MSE for individual ensemble members with shaded regions corresponding to $\pm$ one standard deviation, dashed lines refer to the ensemble prediction. Shown are evaluations on the training- (left), test- (middle) and out of distribution data (right). The lower row shows zoom-ins on the ensemble predictions.

The results of our experiments on the cross-product are shown in Figure 10. As above, we evaluate the orbit MSE on each ensemble member individually (solid lines and shaded region corresponding to $\pm$ one standard deviation) and for the ensemble output (dashed lines). This is true on training- test and out of distribution data. Also in this equivariant task is the ensemble mean about an order of magnitude more equivariant than the ensemble members. As expected from our theory, the ensemble becomes more equivariant for larger ensembles and wider networks.

## D    HISTOLOGICAL SLICES

**Training**    The NCT-CRC-HE-100K dataset Kather et al. (2018) comprises 100k stained histological images in nine classes. In order to make the task more challenging, we only use 10k randomly selected samples, train on 11/12[th] of this subset and validate on the remaining 1/12[th]. We trained ensembles of CNNs with six convolutional layers of kernel size 3 and 6, 16, 26, 36, 46 and 56 output channels, followed by a kernel size 2, stride 2 max pooling operation and three fully connected layers of 120, 84 and 9 output channels. The models had 123k parameters each. We trained the ensembles with the Adam optimizer using a learning rate of 0.001 on batches of size 16. In our training setup, ensemble members reach a validation accuracy of about 96% after 20 epochs, cf. Figure 12.

**Invariance on in-distribution data**    As for our experiments on FashionMNIST, we verify that the ensemble is more invariant as a function of its input than the ensemble members. On training- and validation data this is to be expected since the ensemble predictions have a higher accuracy than the predictions of individual ensemble members. The invariance results on validation data are depicted in Figure 11.

**OOD data**    In order to arrive at a sample of OOD data on which the network makes non-constant predictions, we optimize the input of the untrained ensemble using the Adam optimizer to yield predictions of high confidence ($> 99\%$), starting from 100 random normalized images for each class. We optimize only the $5 \times 5$ lowest frequencies in the Fourier domain to obtain samples which can be rotated without large interpolation losses, yielding samples as depicted in Figure 13.

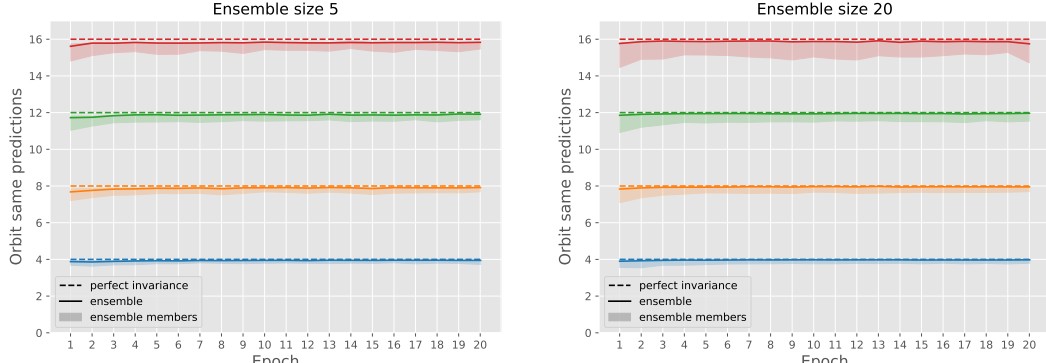

Figure 11: Ensemble invariance on validation data for ensembles trained on histological data. Number of validation samples with the same prediction across a symmetry orbit for group orders 4 (blue), 8 (orange), 12 (green) and 16 (red) versus training epoch for ensemble sizes 5 (left) and 20 (right). The ensemble predictions (solid line) are more invariant than the ensemble members (shaded region corresponding to $25^{th}$ to $75^{th}$ percentile of ensemble members). The effect is larger for ensemble size 20 (right).

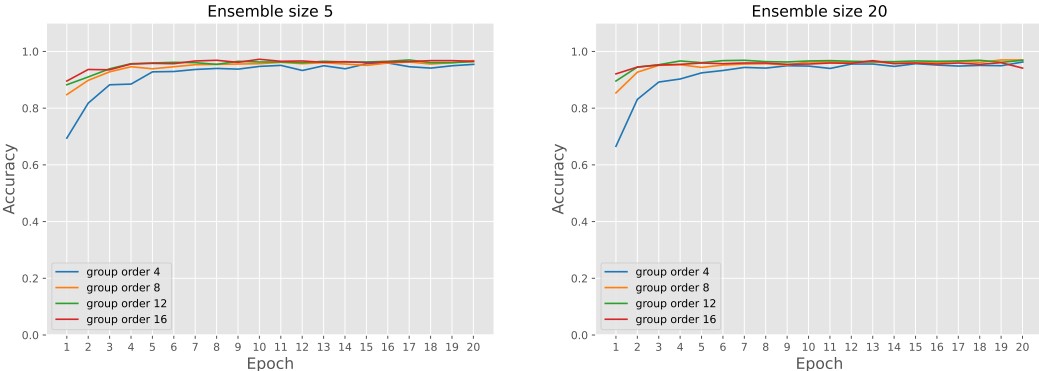

Figure 12: Validation accuracy versus training time for ensemble of size 5 (left) and 20 (right) trained on histological data.

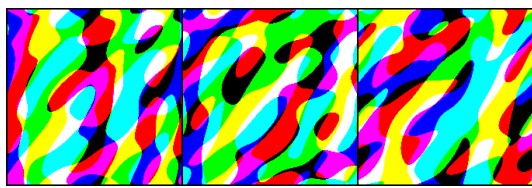

Figure 13: Three OOD data samples for the histology ensemble.

