# OpenReview forum: "Emergence of Equivariance in Deep Ensembles"
_ICLR.cc/2024/Conference — Submitted to ICLR 2024_

### Official Review · Reviewer_waBb · 2023-10-31

**Soundness:** 3 good
**Presentation:** 3 good
**Contribution:** 2 fair
**Rating:** 5
**Confidence:** 4

**Summary:**

This paper proves that an infinite ensemble of neural networks becomes equivariant with data augmentation under mild assumptions. They use neural tangent kernels to show the equivariance. This property is also empirically evaluated with three tasks such as rotated image classification.

**Strengths:**

The paper theoretically shows that the equivariance emerges by model ensemble without hand-crafted architecture design. This direction, obtaining equivariance while we can freely choose networks, is important in practical usage.

Although the paper is theory-flavored, it is easy to read and follow.

**Weaknesses:**

The main finding (emergence of equivariance in deep ensembles) is not very surprising. Data augmentation imposes a bias on a model toward invariance/equivariance, and for me, it's natural to see the averaged model archives that property. I mean, if we have an infinite number of data instances and the model capacity is large enough, the model trained for many steps would be equivariant. The neural tangent approach is of course different from this asymptotic approach, but the main idea should be the same.

The experiments have room for improvement.
1. Instead of equivariance, invariance is evaluated.
2. Only a cyclic group is considered so it is not clear what kind of consequence can we get for more complex groups such as SO(2), SO(3), or SE(3).
3. No comparison with equivariant networks such as steerable CNNs.

**Questions:**

In Equation (13) you assume that we can get the index permutation. However, for continuous groups such as SO(2) we cannot do this. Can you generalize the entire theory to avoid this issue?

---

> ### Author Response · Authors · 2023-11-22
> **Thank you for your review**
>
> > **Strengths:**
> The paper theoretically shows that the equivariance emerges by model ensemble without hand-crafted architecture design. This direction, obtaining equivariance while we can freely choose networks, is important in practical usage.
> Although the paper is theory-flavored, it is easy to read and follow.
> >
>
> We thank the reviewer for the positive feedback.
>
> > **Weaknesses:**
> The main finding (emergence of equivariance in deep ensembles) is not very surprising. Data augmentation imposes a bias on a model toward invariance/equivariance, and for me, it's natural to see the averaged model archives that property. I mean, if we have an infinite number of data instances and the model capacity is large enough, the model trained for many steps would be equivariant. The neural tangent approach is of course different from this asymptotic approach, but the main idea should be the same.
> >
>
> It is indeed intuitive that the ensemble becomes equivariant on the training- and test data when trained with data augmentation. However, our results go significantly beyond this intuition in several important directions:
>
> 1. Our theoretical analysis shows that for large-width ensembles, the equivariance is not only approximate, as expected from data augmentation, but indeed exact.
> 2. For training with data augmentation, one would expect (approximate) equivariance to hold only on the data manifold, but not away from it since the ensemble is only trained on the data manifold. This would be true even in the asymptotic case of infinite training data and -time. However, both our neural tangent kernel based arguments and our experiments show that the ensemble becomes a truly equivariant function *which is also equivariant away from the data manifold*.
> 3. *We show that the ensemble is equivariant for all training times.* This is in contrast to the above intuition which would suggest that the ensemble becomes more and more equivariant throughout training as it learns to fit the augmented training data.
>
> > The experiments have room for improvement.
> >1. Instead of equivariance, invariance is evaluated.
> >
>
> In the new version of the manuscript, we have extended our experiments to also include an equivariant task, see Appendix C and in particular Figure 10. We have trained ensembles of fully-connected networks to predict the cross product in $\mathbb{R}^3$, a task which is equivariant with respect to rotations in SO(3). Our experiments show that the predictions of even small ensembles are significantly more equivariant than the predictions of their ensemble members. This is the case even though we only augment with a small finite subset of the continuous symmetry group and holds on- as well as off manifold.
>
> > 2. Only a cyclic group is considered so it is not clear what kind of consequence can we get for more complex groups such as SO(2), SO(3), or SE(3).
> >
>
> In order to address this point, we have extended our experiments on FashionMNIST. As detailed in Section B.2 in the appendix of the revised manuscript, we have measured invariance with respect to the full SO(2) group of ensembles trained on finite subgroups $C_4$, $C_8$ and $C_6$. Our experiments summarized in Figure 5 show that even for moderately-sized subgroups, the invariance of the ensemble is very high, in accordance to Lemma 6. Furthermore, the ensemble output is much more invariant than individual ensemble members, as is the case for invariance with respect to finite subgroups.
>
> Note that our new experiments on the cross product concern the continuous symmetry group SO(3).

---

> > ### Author Response · Authors · 2023-11-22
> >
> > > 3. No comparison with equivariant networks such as steerable CNNs.
> > >
> >
> > We actually also performed experiments with ensembles of [ESCNNs](https://github.com/QUVA-Lab/escnn) invariant with respect to $C_4$ rotations trained on the FashionMNIST task. As expected, the ensemble members show perfect invariance. For different model- and ensemble sizes, the accuracies reached by the invariant model were
> >
> > | Parameters per model | Ensemble size | Validation accuracy |
> > | --- | --- | --- |
> > | 51k | 10 | 85.3% |
> > | 184k | 10 | 86.0% |
> > | 184k | 40 | 86.3% |
> >
> > Larger ensembles exhausted the VRAM of the GPU we used. For comparison, an ensemble of 100 (non-invariant) CNNs with 120k parameters per ensemble member reached 92.9% validation accuracy.
> >
> > However, a comparison between the ESCNNs and the CNNs is complicated by the different parameter counts of the architectures and the necessity for a large spatial pooling layer in the invariant models, which is not present in the non-invariant models. We therefore decided to not include these experiments in the final version of the manuscript.
> >
> > Furthermore, we want to stress that our manuscript is a theory paper. Its main objective is to theoretically elucidate surprising emergent capabilities of deep ensembles by using neural tangent kernel theory. We did not aim to propose a highly competitive method which would probably involve applying several tricks of the trade which are harder to model theoretically, such as sampled data augmentation as well as non-iid ensemble member selection.
> >
> > > **Questions:**
> > In Equation (13) you assume that we can get the index permutation. However, for continuous groups such as SO(2) we cannot do this. Can you generalize the entire theory to avoid this issue?
> > >
> >
> > This question is addressed by Lemma 6 and the additional experiments of the revised Appendix B.2.
> >
> > Full data augmentation for continuous groups would require infinite training data. For finite training data, augmentation transformations form a finite subset of the continuous group.
> >
> > This is the case studied theoretically in Lemma 6, where we provide a bound on the invariance error of an ensemble augmented with finitely many transformations from a continuous symmetry group. We show that the invariance error approaches zero as the discretization improves and more closely approximates the full group.
> >
> > Experimentally, we show in Figure 5 of Appendix B.2 that emergent equivariance of deep ensembles also extends to the continuous SO(2) symmetry, see discussion above.

---

### Official Review · Reviewer_8JsL · 2023-11-01

**Soundness:** 3 good
**Presentation:** 4 excellent
**Contribution:** 3 good
**Rating:** 8
**Confidence:** 3

**Summary:**

In this paper, the authors prove that when trained with data augmentation, deep ensembles are equivariant during training. They use the theory of NTKs and explicitly prove that the deep ensembles are equivariant regardless of the training step or data. However, this is limited by the fact that ensembles are finite, networks are not infinitely wide, and there is a limit to data augmentation for continuous groups. The authors further provide error bounds considering these limitations.

**Strengths:**

Although I am not very familiar with neural tangent kernels, the authors presented the work in a way that was easy to follow. Theorem 4 in particular seems like a very strong result. The authors further consider practical and very relevant limitations such as the finite ensemble, continuous groups, and finite width and prove error bounds. The experiments support the theory.

**Weaknesses:**

The type of data augmentation considered seems perhaps a little strong. By using all elements of the group orbit, it naturally lends itself to rewriting the group action as permutations, which seems to be critical in the proof. However, many common data augmentation strategies involve loss of information (e.g. random crops, random non-circular shifts, etc.). If the authors could provide any insights or foreseeable limitations of this work for other data augmentation types, that would be very helpful.

**Questions:**

See weaknesses

---

> ### Author Response · Authors · 2023-11-22
> **Thank you for your review**
>
> > **Strengths:**
> Although I am not very familiar with neural tangent kernels, the authors presented the work in a way that was easy to follow. Theorem 4 in particular seems like a very strong result. The authors further consider practical and very relevant limitations such as the finite ensemble, continuous groups, and finite width and prove error bounds. The experiments support the theory.
> >
>
> We thank the reviewer for the positive feedback.
>
> > **Weaknesses:**
> The type of data augmentation considered seems perhaps a little strong. By using all elements of the group orbit, it naturally lends itself to rewriting the group action as permutations, which seems to be critical in the proof. However, many common data augmentation strategies involve loss of information (e.g. random crops, random non-circular shifts, etc.). If the authors could provide any insights or foreseeable limitations of this work for other data augmentation types, that would be very helpful.
> >
>
> Although many commonly used data augmentation strategies involve a loss of information and are therefore not strictly speaking symmetry transformations, they can usually be interpreted as approximating some group transformation. E.g. random crops approximate scalings and non-circular shifts approximate translations. In particular for network architectures which use local transformations of the input features like convolutions or windowed self-attention, the deviations from a strict symmetry transformation are often restricted to some regions of the input, e.g. the edges. On these grounds, we expect the observed effects of emerging equivariance for ensembles trained with data augmentation to hold approximately in the case of lossy augmentation strategies.
>
> The approximation in going from an augmentation involving information loss to an augmentation which is an exact symmetry transformation is comparable to augmenting with a finite subgroup of a continuous symmetry group and evaluating equivariance for the full symmetry group. For this case, we provide a theoretical bound on the equivariance error in Lemma 6. As our new experiments (see Appendix B.2, and in particular Figure 5, of the revised manuscript) on the FashionMNIST dataset in this setting demonstrate, shows the ensemble small equivariance errors on the full symmetry group, even when trained on relatively small finite subgroups. Therefore, we expect a similar emergence of true equivariance when going from a lossy data augmentation scheme to one which involves exact symmetry transformations.

---

> > ### Comment · Reviewer_8JsL · 2023-11-23
> >
> > I thank the authors and am satisfied with the response. I maintain my original score.

---

### Official Review · Reviewer_FAEm · 2023-11-02

**Soundness:** 3 good
**Presentation:** 3 good
**Contribution:** 3 good
**Rating:** 6
**Confidence:** 2

**Summary:**

This work considers deep ensembles in the infinite width limit / NTK regime. For a deep ensemble on a dataset with equivariant data augmentation to a symmetry group, the work shows that the deep ensemble is equivariant at all points in its training evolution. Bounds are given on the behavior of different approximations to this, in the cases of: finite ensembles and finite subgroups for data augmentation. Empirical results show that numerically trained ensembles approach equivariance as width or number of models in the ensemble increase.

**Strengths:**

1. Well-written and well-organized Section 5: the proof sketch is nice.
2. Interesting empirical results that support the theory. The ensembles become more equivariant as width or number of models increases.

**Weaknesses:**

1. Do these results prescribe any particular practical methods, or does it give particular insights on models trained in practice? There does not seem to be much discussion on this. For instance, do people often train ensembles on equivariant data, and how does this compare to single models?
2. Could use more details on the critical assumption on the input layer, see question 1 below.

**Questions:**

1. Does your assumption on the networks depending on input through $w^{(k)}x$ on Page 5 really hold for CNNs? CNNs have their filter coefficients initialized via centered Gaussians, but the underlying matrix is not (because of weight sharing). Thus, there are orthogonal transformations on the input that may change the output (e.g. permute top left with top right pixel).
2. Intuitively, what does the deep ensemble output look like at initialization? I am trying to intuit why it is equivariant then.
3. Could you give more explanation or intuition about $C(x)$ in Lemma 6 in the main text?

---

> ### Author Response · Authors · 2023-11-22
> **Thank you for your review**
>
> > **Strengths:**
> >1. Well-written and well-organized Section 5: the proof sketch is nice.
> >2. Interesting empirical results that support the theory. The ensembles become more equivariant as width or number of models increases.
> >
>
> We thank the reviewer for the positive feedback.
>
> > **Weaknesses:**
> >1. Do these results prescribe any particular practical methods, or does it give particular insights on models trained in practice? There does not seem to be much discussion on this. For instance, do people often train ensembles on equivariant data, and how does this compare to single models?
> >
>
> In our view, emergent equivariance of deep ensembles is mainly interesting in scenarios for which ensembles are also used for different reasons such as uncertainty prediction and robustness. It is an important contribution of our manuscript to prove that equivariance comes for free in these scenarios. We stress that deep ensembles are widely used and thus this is a finding of immediate practical value. For example, in protein structure prediction, many models use an ensemble to estimate the uncertainty of the prediction, see e.g. [1, 2]. There are even some works in this context that rely on non-equivariant architectures and ensure equivariance by averaging over an appropriate subset of the group orbit [3]. It is an exciting direction of future research to harness our insights to impose equivariance by ensemble. Note that we demonstrate in the revised Appendices B.2 and C that deep ensembles can also lead to emergent SO(2) and SO(3) equivariance, respectively.
>
> [1] Ruffolo, Jeffrey A., et al. "Fast, accurate antibody structure prediction from deep learning on massive set of natural antibodies." *Nature communications* 14.1 (2023): 2389. [https://www.nature.com/articles/s41467-023-38063-x](https://www.nature.com/articles/s41467-023-38063-x)
>
> [2] Abanades, Brennan, et al. "ImmuneBuilder: Deep-Learning models for predicting the structures of immune proteins." *Communications Biology* 6.1 (2023): 575. [https://www.nature.com/articles/s42003-023-04927-7](https://www.nature.com/articles/s42003-023-04927-7)
>
> [3] Martinkus, Karolis, et al. "AbDiffuser: full-atom generation of in-vitro functioning antibodies." NeurIPS 2023, *arXiv preprint arXiv:2308.05027* (2023).
>
> > 2. Could use more details on the critical assumption on the input layer, see question 1 below.
> >
>
> See below
>
> > **Questions:**
> >1. Does your assumption on the networks depending on input through $w^k x$ on Page 5 really hold for CNNs? CNNs have their filter coefficients initialized via centered Gaussians, but the underlying matrix is not (because of weight sharing). Thus, there are orthogonal transformations on the input that may change the output (e.g. permute top left with top right pixel).
> >
>
> It is true that there are orthogonal transformations to a CNN which change its output. However, in the NTK, we are concerned with the expectation value over the (inner product of) derivatives of the output with respect to the trainable parameters. If we perform an orthogonal transformation of the input domain, the derivatives change, but their expectation value over initializations remains the same since the initialization distribution is the same for all filter components.
>
> > 2. Intuitively, what does the deep ensemble output look like at initialization? I am trying to intuit why it is equivariant then.
> >
>
> The ensembles output at initialization is constant, see Eq. 6. As a result, the network is trivially equivariant for all symmetry groups. Data augmentation ensures that this equivariance is not broken by training although the output is no longer constant.
>
> > 3. Could you give more explanation or intuition about C(x) in Lemma 6 in the main text?
> >
>
> There are some aspects of C that can indeed be intuited:
> - The constant C vanishes at the beginning of training as the ensemble is trivially equivariant due to its constant output.
> - The constant depends on an expectation over initializations involving the Libschitz constant as well as the expected gradient of the network over initializations. This is to be expected since violations of equivariance with respect to the continuous symmetry will strongly depend on how drastically the ensemble members can change between the points on the group orbit covered by the discretization of the continuous symmetry.

---

### Official Review · Reviewer_wXhZ · 2023-11-03

**Soundness:** 3 good
**Presentation:** 3 good
**Contribution:** 2 fair
**Rating:** 5
**Confidence:** 4

**Summary:**

This paper shows how, in a large width limit and with the inclusion of data augmentation, a generic deep ensemble becomes inherently equivariant. This equivariance is observed at each training step, irrespective of the chosen architecture, contingent upon the utilization of data augmentation. Notably, this equivariance extends beyond the observed data manifold and emerges through the collective predictions of the ensemble, even though individual ensemble member is not equivariant. It provides both theoretical proof, utilizing neural tangent kernel theory, and experiments to support and validate these observations.

**Strengths:**

1) This paper presents a very interesting idea of the emergence of equivariance with data augmentation and model ensembles.

2) This paper is generally well-written.

3) The theoretical claims in the paper are sound.

**Weaknesses:**

1) This paper lacks a proper comparison with other methods that can bring equivariance without any constraint on the architecture like [1, 2, 3, 4, 5]. When showing the out-of-distribution transformation results it'll be great to compare with those methods. The current results in the paper are more like ablations of the proposed augmentation and ensembling technique. It is not clear where it stands with other architecture-agnostic equivariance methods. (even if the proposed method does poorly compared to those it'll be good to have those results)

2) The author claims data augmentation is the only alternate method to bring equivariance in a non-equivariant model. I'll refer these papers [1,5] to the authors where they show that equivariance can be achieved using symmetrization and canonicalization. It'll be nice to include those as well in the paper. Especially symmetrization is closely related to the idea of ensembling because you pass different transformations of the same image throughout the same network before you average. My intuition is that symmetrization keeps the architecture the same and transforms the input, whereas the current work keeps the input the same and learns a transformer version of weights or each of the networks learning to process different transformations of the input. It'll be great if the authors can shed some light on the connection and discuss architecture agnostic body of work.


[1] Puny, O., Atzmon, M., Ben-Hamu, H., Misra, I., Grover, A., Smith, E. J., & Lipman, Y. (2021). Frame averaging for invariant and equivariant network design. arXiv preprint arXiv:2110.03336.

[2] Mondal, A. K., Panigrahi, S. S., Kaba, S. O., Rajeswar, S., & Ravanbakhsh, S. (2023). Equivariant Adaptation of Large Pre-Trained Models. arXiv preprint arXiv:2310.01647.

[3] Basu, S., Sattigeri, P., Ramamurthy, K. N., Chenthamarakshan, V., Varshney, K. R., Varshney, L. R., & Das, P. (2023, June). Equi-tuning: Group equivariant fine-tuning of pretrained models. In Proceedings of the AAAI Conference on Artificial Intelligence (Vol. 37, No. 6, pp. 6788-6796).

[4] Basu, Sourya, et al. "Equivariant Few-Shot Learning from Pretrained Models." arXiv preprint arXiv:2305.09900 (2023).

[5] Kaba, S. O., Mondal, A. K., Zhang, Y., Bengio, Y., & Ravanbakhsh, S. (2023, July). Equivariance with learned canonicalization functions. In International Conference on Machine Learning (pp. 15546-15566). PMLR.

**Questions:**

See the weaknesses above

---

> ### Author Response · Authors · 2023-11-22
> **Thank you for your review**
>
> ### **Strengths:**
>
> **1. This paper presents a very interesting idea of the emergence of equivariance with data augmentation and model ensembles.
> 2. This paper is generally well-written.
> 3. The theoretical claims in the paper are sound.**
>
> We thank the reviewer for the positive feedback.
>
> ### **Weaknesses:**
>
> 1. **This paper lacks a proper comparison with other methods that can bring equivariance without any constraint on the architecture like [1, 2, 3, 4, 5]. When showing the out-of-distribution transformation results it'll be great to compare with those methods. The current results in the paper are more like ablations of the proposed augmentation and ensembling technique. It is not clear where it stands with other architecture-agnostic equivariance methods. (even if the proposed method does poorly compared to those it'll be good to have those results)**
>
> In light of your comments, we have substantially expanded the related works section discussing similarities and differences to the suggested references (see point immediately below).
>
> Furthermore, we have added additional numerical experiments in Appendix B.2 comparing deep ensembles to the canoncialization method of [5]. Briefly summarized, we find the following: We use the exact same model architecture for both the predictor of the canonicalization and the members of the deep ensemble and train on FashionMNIST. Canonicalization leads to an exactly equivariant model whereas our method yields approximate equivariance due to finite number of ensemble members and finite width. As a result, a comparison in the orbit same prediction metric is trivial and clearly shows the benefits of the canonicalization approach. On the other hand, deep ensembles have the advantage that they naturally benefit from well-known advantages of deep ensembles, such as improved accuracy and robustness as well as natural uncertainty estimation. Interestingly, we find a comparable scaling of equivariance to the full SO(2) group when we compare deep ensembles trained with $C_k$ data augmentation with the corresponding canonicalized $C_k$ models, see Figure 5 and 6. Furthermore, we find that deep ensembles lead to a validation accuracy of 91% while the canonicalized model reaches 87%. Both comparisons however come with important caveats: canonicalization allows for full SO(2) equivariance by choosing an appropriate canonicalizer. Futhermore, we are comparing an ensemble of models with a single canonalized model (with the same architecture). An ensembling of canonalized models most likely closes the performance gap.
>
> In our view, emergent equivariance of deep ensembles is particularly interesting in scenarios for which ensembles are also used for different reasons such as uncertainty prediction and robustness. It is an important contribution of our manuscript to prove that an approximate form of equivariance comes for free in these scenarios. We stress that deep ensembles are widely used and thus this is a finding of immediate practical value.
>
> Finally, we want to highlight that our manuscript is a theory paper. Its main objective is to theoretically elucidate surprising emergent capabilities of deep ensembles by using neural tangent kernel theory. We did not aim to propose a highly competitive method which would likely involve applying several tricks of the trade which are harder to model theoretically, such as sampled data augmentation as well as non-iid ensemble member selection.

---

> > ### Author Response · Authors · 2023-11-22
> >
> > 1. **The author claims data augmentation is the only alternate method to bring equivariance in a non-equivariant model. I'll refer these papers [1,5] to the authors where they show that equivariance can be achieved using symmetrization and canonicalization. It'll be nice to include those as well in the paper. Especially symmetrization is closely related to the idea of ensembling because you pass different transformations of the same image throughout the same network before you average. My intuition is that symmetrization keeps the architecture the same and transforms the input, whereas the current work keeps the input the same and learns a transformer version of weights or each of the networks learning to process different transformations of the input. It'll be great if the authors can shed some light on the connection and discuss architecture agnostic body of work.**
> >
> > **[1] Puny, O., Atzmon, M., Ben-Hamu, H., Misra, I., Grover, A., Smith, E. J., & Lipman, Y. (2021). Frame averaging for invariant and equivariant network design. arXiv preprint arXiv:2110.03336.**
> >
> > **[2] Mondal, A. K., Panigrahi, S. S., Kaba, S. O., Rajeswar, S., & Ravanbakhsh, S. (2023). Equivariant Adaptation of Large Pre-Trained Models. arXiv preprint arXiv:2310.01647.**
> >
> > **[3] Basu, S., Sattigeri, P., Ramamurthy, K. N., Chenthamarakshan, V., Varshney, K. R., Varshney, L. R., & Das, P. (2023, June). Equi-tuning: Group equivariant fine-tuning of pretrained models. In Proceedings of the AAAI Conference on Artificial Intelligence (Vol. 37, No. 6, pp. 6788-6796).**
> >
> > **[4] Basu, Sourya, et al. "Equivariant Few-Shot Learning from Pretrained Models." arXiv preprint arXiv:2305.09900 (2023).**
> >
> > **[5] Kaba, S. O., Mondal, A. K., Zhang, Y., Bengio, Y., & Ravanbakhsh, S. (2023, July). Equivariance with learned canonicalization functions. In International Conference on Machine Learning (pp. 15546-15566). PMLR.**
> >
> > We agree that this is an interesting connection that we discuss in detail in the revised manuscript.  Thank you for pointing this out.
> >
> > We also largely share your intuition on the connection between deep ensembles and symmetrization. The latter applies the same neural network to all members of the group orbit (or in the case of [1] a cleverly chosen input-dependent subset) during inference. Deep ensembles do not need to transform the input for inference. In contrast, they take the average over several models trained with data augmentation. Note however that the emergent equivariance of deep ensembles is not solely a consequence of training as the ensemble is also equivariant *off the data manifold and at any training time t*. Rather, there are two effects that complement each other: i)at the beginning of training the ensemble is equivariant as the output is constant ii) data augmentation ensures that the equivariance is not broken by training updates.
> >
> > An interesting consequence of this is that if a symmetry is not known but present in the data, the deep ensemble will automatically be equivariant with respect to it. For symmetrization, this would not be the case as one would need to sum over the corresponding group orbit. On the other hand, symmetrization has the advantage that the resulting model is exactly equivariant while the equivariance of deep ensembles only becomes exact in the infinite width limit.

---

### Author Response · Authors · 2023-11-22
**Overview**

We thank the reviewers for the valuable feedback.

All reviewers agreed that the emergent equivariance of deep ensemble is an interesting result and appreciated both our theoretical derivations of it as well as our experiments.

We have incorporated the suggestions by the reviewers in the revised manuscript. For convenience, we have marked all changes to the manuscript in blue color.  Briefly summarize the changes are as follows:

- We added an extensive discussion of how emergent equivariance of deep ensembles compares to symmetrization and canonicalization in the related works section. These frameworks also ensure equivariance without any constraints on the model.
- We added additional experiments:
    - In Appendix B.2 was revised with additonal experiments which demonstrate that emergent equivariance extends to continuous SO(2) symmetries, as anticipated by Lemma 6 and a comparison to canonicalization.
    - We added experiments in Appendix C for an equivariant problem setting as opposed to an invariant one.

We hope we have addressed the questions by the reviewers and incorporated all their feedback. Please let us know if you have any further comments or questions.

---

### Meta-Review · Area_Chair_YTER · 2023-12-06

**Metareview:**

This paper makes a theoretical argument that - in the inifinite-width and infinite-ensemble-member limit, deep ensembles become equivariant with respect to any cyclic group. The phenomenon investigated by the authors is novel, timely, and likely of large interest to the neural network community. At the same time, there are concerns that (1) there is not a sufficient comparison to other equivariant baselines (which suggests that there may be limitations to this paper’s practicality) and (2) that the authors’ claim of equivariance away from the training data manifold may be too general than what can be claimed by their results. I believe that this work can eventually be of value to the community, but it could benefit from revision and further reviewing.

Metareviewer note: I have read the paper myself, and I would like to bring up two additional notes for the authors:

1) Equation 8 only holds when the loss considered is the MSE loss, as well as when the scaling/learning rate meet the NTK conditions. The authors should make this explicit.

2) The derivation of equation 3 brings up many questions that are not answered by the paper. An infinite-ensemble of infinitely-wide neural networks has two limits (width and number of ensemble members), and so care is required to ensure that these limits resolve jointly. What is the order of the limits that the authors are considering? The authors should include a derivation of this joint limit to prove its validity.

**Justification For Why Not Higher Score:**

This paper has outstanding concerns about (1) experiments and (2) the scope of the claims. After the reviewing and discussion period, I thoroughly read the paper myself and found that these issues (and others) were present and had not been fully addressed by the authors. Therefore, I believe this paper would benefit from revisions and further reviewing.

**Justification For Why Not Lower Score:**

N/A

---

### Decision · Program_Chairs · 2024-01-16

Reject